# LookWhere? Efficient Visual Recognition by Learning Where to Look and What to See from Self-Supervision

• **Anthony Fuller**[1,3]    • **Yousef Yassin**[1]
**Junfeng Wen**[1]    **Daniel G. Kyrollos**[1]    **Tarek Ibrahim**[1]
⋆ **James R. Green**[1]    ⋆ **Evan Shelhamer**[1,2,3]
Carleton University[1]    University of British Columbia[2]    Vector Institute[3]
• Co-first author    ⋆ Co-advising author

## Abstract

Vision transformers are ever larger, more accurate, and more expensive to compute. The expense is even more extreme at high resolution as the number of tokens grows quadratically with the image size. We turn to adaptive computation to cope with this cost by learning to predict *where* to compute. Our LookWhere method divides the computation between a low-resolution selector and a high-resolution extractor *without ever processing the full high-resolution input*. We jointly pretrain the selector and extractor *without task supervision* by distillation from a self-supervised teacher, in effect, learning where and what to compute simultaneously. Unlike prior token reduction methods, which pay to save by pruning already-computed tokens, and prior token selection methods, which require complex and expensive per-task optimization, LookWhere economically and accurately selects and extracts transferrable representations of images. We show that LookWhere excels at sparse recognition on high-resolution inputs (Traffic Signs), maintaining accuracy while reducing FLOPs by up to $34\times$ and time by $6\times$. It also excels at standard recognition tasks that are global (ImageNet classification) or local (ADE20K segmentation), improving accuracy while reducing time by $1.36\times$. See `https://github.com/antofuller/lookwhere` for the code and weights.

## 1   Introduction: A Look through Self-Supervised Eyes

Self-supervised computer vision models can identify informative inputs, as shown in Fig. 1, by **what** DINOv2 [1] attends to and **where**, and without any knowledge of the task to be done. In short, these methods solve their own learning problems: given one view of visual data, predict another (by reconstructive [2, 3, 4, 5, 6] or contrastive learning [1, 7, 8, 9]). When learned at scale, they attend to the visually interesting by identifying what is informative for their predictions; importantly, this input may be sparser than the full input, especially at high resolution, offering an opportunity for efficiency.

In this work, we reduce the computation for recognition by *predicting* this visual interest without *fully processing* the input. We rely on attention and representation in the eyes of the self-supervised model to learn both where to look, with an efficient **selector** of locations, and what to see, with an expressive **extractor** of representations. To do so, we propose **LookWhere**: a framework for self-supervised adaptive computation that factorizes where and what, for learning and inference, with paired selector-extractor models. Our selector and extractor together represent a high-res input given ❶ a **low-res** version and ❷ a sparse set of **high-res** patches selected from the high-res version. Our novel selector-extractor architecture and what-where distillation training effectively approximate deep self-supervised representations with efficient adaptive predictions.

LookWhere applies to imagery of any size, but can help most at high resolution. High resolution (with pixel dimensions of $>1000$ on a side) is now common in photography [10], remote sensing

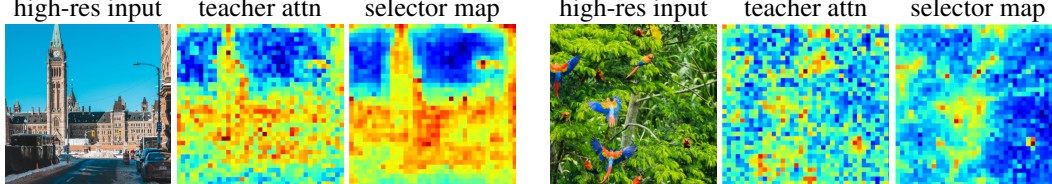

| high-res input | teacher attn | selector map | high-res input | teacher attn | selector map |

Figure 1: **Attention as Supervision**. Self-supervised models (like DINOv2 [1]) learn *what* is visually interesting without tasks or labels. We use a last step of their deep computation, the final attention map, to start ours; supervising an efficient selector that predicts *where* to process for adaptive computation.

[11], autonomous driving [12, 13], medical imaging [14, 15], and more. These high-resolution inputs offer *detail* and *context* to improve recognition [16, 17, 18, 19], but high-res computation is costly.

This cost has led to active research on adaptive computation for efficiency. These works, and ours, are motivated by the intense computation of vision transformers/ViTs [20, 21], and the opportunity in their structure: by first converting an image into tokens, then only applying pair-wise attention and token-wise FFN operations, the amount of computation scales directly with the number of tokens (quadratically for pair-wise and linearly for token-wise). Each token eliminated is efficiency gained.

LookWhere improves on existing methods for token reduction [22, 23, 24, 25, 26, 27], and token selection [28, 29, 30, 31] in its efficiency and simplicity. Unlike token reduction methods, it never processes all input tokens, which saves computation (especially on high-res inputs). Unlike prior token selection methods, it is efficient and straightforward to pretrain and finetune. Unlike both, it jointly pretrains its selector-extractor models for transfer across diverse visual recognition tasks.

We show that LookWhere delivers state-of-the-art (SoTA) accuracy/efficiency trade-offs in comparisons and controlled experiments across tasks and resolutions. We measure its efficiency during testing and training, and find it achieves these results with the least per-task training computation. We first demonstrate LookWhere in standard benchmarks (ImageNet [32], ADE20K [33]) to prove its effectiveness. Then, we demonstrate LookWhere for high-resolution data on an established special-purpose benchmark (Traffic Signs [34]) to emphasize when adaptive computation is needed.

## 2 Method: Selector-Extractor Computation and What-Where Distillation

LookWhere accelerates inference by approximating full, deep representations with adaptive computation of predictions learned from distillation. The selector predicts where to look, the extractor predicts what to see, and each learns by distillation from the attention and tokens of a self-supervised teacher (Fig. 2). We call this novel joint learning scheme **what**-**where** distillation.

The selector and extractor work in tandem to efficiently extract task-specific representations, given minimal finetuning, based on their self-supervised pretraining of what and where to compute. We first explain inference with the selector and extractor models and their computation (Sec. 2.1). Next, we explain finetuning for a task (Sec. 2.2). Last, we explain pretraining (Sec. 2.3), which is needed just once for a visual domain to enable fast task-wise finetuning and deployment.

We review ViT computation and notation as a prerequisite for our adaptive computation of ViTs. This brief summary clarifies the computations that LookWhere approximates and accelerates.

**Patchification.** To process an image of dimensions $R \times R$ with $C$ channels, ViTs split the image into an $N \times N$ grid of patches $\mathbb{R}^{R \times R \times C} \to \mathbb{R}^{N \times N \times P \times P \times C}$ where $P$ is the resolution of a patch. This grid is then flattened into a sequence of $N^2$ patches, each of which is linearly projected to a $D$-dimensional embedding called a *token* $\mathbb{R}^{N \times N \times P \times P \times C} \to \mathbb{R}^{N^2 \times D}$. The patch tokens $x_{pat} \in \mathbb{R}^{N^2 \times D}$ are now ready for attention and feedforward network (FFN) computation.

**Computation.** ViTs process the tokens like a generic transformer: through a stack of architecturally-identical layers composed of pair-wise self-attention and token-wise FFN operations. The computational cost scales with tokens: self-attention scales quadratically and the FFN scales linearly.

**Attention.** Self-attention applies three learned linear projections to each token in the sequence, producing query, key, and value vectors. The dot product similarity between queries, and keys is the "attention" for the pair. The output for each query is the weighted sum of the values given by the

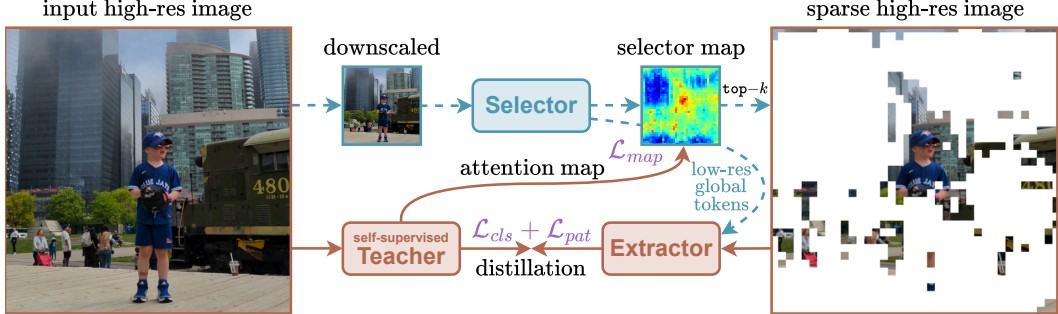

input high-res image · downscaled · selector map · sparse high-res image

Selector · top−$k$ · $\mathcal{L}_{map}$ · attention map · low-res global tokens

self-supervised Teacher · $\mathcal{L}_{cls} + \mathcal{L}_{pat}$ · distillation · Extractor

Figure 2: **LookWhere is trained by distillation** from a self-supervised teacher to learn *where* to compute, from the attention map, and *what* to compute, from the class and patch token representations. Only the teacher sees the high-res input to make its attention and tokens for the losses. The selector predicts *where* to look from the low-res input. The extractor predicts *what* to see (e.g. a boy in a hat and jersey, a coffee cup, . . . ) from the top $k$ high-res patches (and low-res tokens) from the selector.

normalized attention for all pairs. (This is normally repeated across $H$ distinct "heads".) Tokens with *lower* attention add *less* to these sums, and raise the potential for efficiency by not computing them.

**Global Tokens.** Special tokens summarize *global* information to complement the local information in the patch tokens. ViTs augment the patch sequence $x_{pat}$ with a learnable "class" token $x_{cls} \in \mathbb{R}^D$, and this technique can extend to multiple "register" tokens $x_{reg} \in \mathbb{R}^{G \times D}$ for $G$ tokens [35].

The full token sequence is $x \triangleq (x_{cls}, x_{reg}, x_{pat}) \in \mathbb{R}^{(1+G+N^2) \times D}$, and its length is dominated by the $N^2$ patch tokens. This is the opportunity for adaptive computation: fewer tokens, less computation.

## 2.1 Selector-Extractor Inference with Split Resolution Computation

LookWhere factorizes inference into a low-resolution *selector* for *where* to compute and a high-resolution *extractor* for *what* to compute. The selector operates on a low-resolution edition of the input to predict the locations of informative patches—producing a *selector map*—and also generates low-resolution summaries of the input. Only the top $k$ most informative patches are selected from the high-resolution input. The extractor processes only these selected patches and re-uses the provided low-resolution summaries to predict a complete representation of the high-resolution input without ever processing all its pixels.

LookWhere admits a variety of architectures for the selector and extractor. Both must be differentiable for end-to-end learning. The selector should be fast in order to efficiently choose and provide the input for the extractor. The extractor should be able to harness spatial sparsity to accelerate its computation of the selected input. We satisfy these considerations by choosing ViTs for both.

The selector's input and depth are reduced for efficient computation. We resize the input to a smaller $R_{\text{low}}$, and truncate the model to its first $L_{\text{low}}$ layers; both modifications compound for higher efficiency. The selector produces latents $z_{\text{low}} \in \mathbb{R}^{(1+G+N^2_{\text{low}}) \times D}$, where the $N^2_{\text{low}}$ patch tokens are mapped by FFN to $N^2_{\text{high}}$ patch importance scores $\mathbb{R}^{N^2_{\text{low}} \times D} \to \mathbb{R}^{N_{\text{high}} \times N_{\text{high}}}$ to yield the selector map $\hat{A}_{\text{high}}$.

For effective extractor computation, we take high-resolution and low-resolution inputs and maintain higher capacity. The top $k$ locations in $\hat{A}_{\text{high}}$ are selected as patches in the high-resolution input. The extractor tokenizes only the selected patches into $x^{pat}_{\text{high}} \in \mathbb{R}^{k \times D}$ and re-uses the selector's low-resolution global tokens $z^{cls}_{\text{low}}$ and $z^{reg}_{\text{low}}$ (to replace the extractor's own $x_{cls}$ and $x_{reg}$ without more computation). Given these high-resolution and low-resolution tokens, the extractor computes its patch token and global token representations following standard ViT inference. We choose the ViT-B architecture for sufficient capacity to represent what is in the input. The extractor outputs latents $\hat{z}_{\text{high}} \in \mathbb{R}^{(1+G+k) \times D}$. For a full representation of the high-resolution input, the sparse set of patch tokens are then spatially interpolated to the high-resolution grid $\hat{z}_{\text{high}} \in \mathbb{R}^{(1+G+N^2_{\text{high}}) \times D}$.

Task-specific modeling finally outputs predictions $\hat{y}$ from $\hat{z}_{\text{high}}$ depending on the details of the task.

## 2.2 Finetuning LookWhere for a Task by Supervised Learning

To apply LookWhere to a given task, we (1) add a task-specific predictor to the extractor and (2) finetune the extractor-predictor by supervised learning. For simplicity, a linear model on the extractor representation suffices to map from its global class token or local patch tokens to the output space. Finetuning LookWhere is highly efficient: we only update the predictor-extractor without updating the selector or needing the teacher. Having already pretrained our selector-extractor to approximate the self-supervised teacher, we can finetune only what they compute (the representation) for the task while sharing where they compute (the adaptive computation) across tasks. The selector further accelerates finetuning: its adaptive computation continues to reduce high-resolution processing at every step. LookWhere is more efficient during inference and during finetuning.

## 2.3 Pretraining by What-Where Distillation of Self-Supervision

Distillation trains a student model to predict a reference output from a teacher model [36]. LookWhere trains its selector and extractor as a pair of students to approximate a deep self-supervised model as the teacher. We distill a self-supervised model for accuracy, efficiency, and transferability. Self-supervised visual representations are now effective for many tasks [1, 5, 7, 37, 38, 39, 40]. Furthermore, their attention maps can identify informative patches in images (per our Fig. 1 and prior analyses [1, 35]). While their computation is expensive, our learned approximation need not be due to adaptive computation and the architectural decoupling of the selector-extractor enabled by distillation.

We propose simultaneous distillation of where and what to compute. We distill where from the teacher attention and what from the teacher representation. The teacher fully processes the high-resolution input to provide the attention and representation for learning. By distilling the teacher's attention, the selector learns to predict the location of deep, salient information given only shallow, low-resolution input. The extractor learns to predict the full representation of the high-resolution input, given only sparse high-resolution tokens and the selector's low-resolution global tokens, by distilling the teacher representation. We pretrain the selector and extractor jointly and in parallel from partial inputs that are more efficient to compute. Note that the selector and extractor learn to predict the representation of an image they never fully see, which is itself a self-supervised task: the masked reconstruction of the representation from only partial inputs and computations. These partial inputs and computations make pretraining more efficient because only the teacher representation is extracted at high resolution.

**Teacher.** We choose DINOv2 [1] as the teacher for its ViT architecture and internet-scale pretraining on diverse images without annotations. We use the variant with $G=4$ register tokens to reduce attention artifacts [35], as we use its attention to train the selector. The teacher processes inputs at resolution $R_{\text{high}}=518$ with patch size $P=14$ for a grid of $N_{\text{high}}=37$ patches. (This is unaltered from DINOv2.) To distill its attention, we extract the unnormalized attention among its patch tokens at the last layer and then average over queries and heads. This approximates where patch tokens contributed to the deepest teacher representation. To distill its representation, we extract the class and patch tokens at the last layer: $z_{\text{high}} \in \mathbb{R}^{(1+N_{\text{high}}^2) \times D}$. We never update the teacher: its parameters are fixed.

**Losses and Updates.** We train LookWhere jointly with three losses for what to compute, by distillation of the class token and patch tokens, and where to compute, by distillation of attention.

- *Class Token Distillation*: We distill the teacher's class token via mean-squared error (MSE): $\mathcal{L}_{cls} = \text{MSE}(\hat{z}_{\text{high}}^{cls}, z_{\text{high}}^{cls})$. This trains the selector and extractor to learn global representations of the high-res input, given the low-res input and selected high-res patches.
- *Patch Token Distillation*: We distill the teacher's patch representations via the mean-squared error (MSE): $\mathcal{L}_{pat} = \text{MSE}(\hat{z}_{\text{high}}^{pat}, z_{\text{high}}^{pat})$. This trains the selector and extractor to learn local representations of the high-res input, given the low-res input and selected high-res patches.
- *Attention Distillation*: We distill the teacher's attention map via the Kullback–Leibler (KL) divergence: $\mathcal{L}_{map} = \text{KL}(\hat{A}_{\text{high}}, A_{\text{high}})$. This trains the selector to predict where the teacher computes.

Our pretraining loss is their sum $\mathcal{L} = \lambda_{cls}\mathcal{L}_{cls} + \lambda_{pat}\mathcal{L}_{pat} + \lambda_{map}\mathcal{L}_{map}$. We set $\lambda_{cls}, \lambda_{pat} = 1$, and $\lambda_{map} = 0.1$. All losses are optimized end-to-end by the extractor and selector. We update the selector and extractor jointly and in parallel without custom batching, balancing, or tuning.

**Initialization.** We set the selector and extractor parameters to those of the teacher: DINOv2.

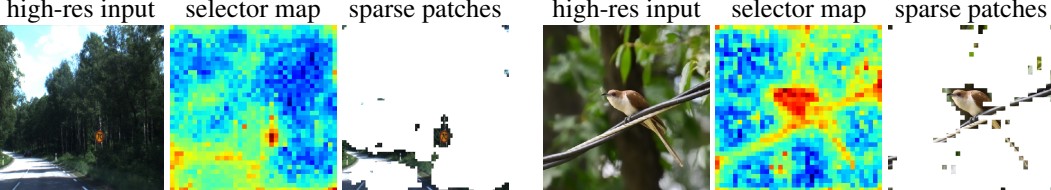

| high-res input | selector map | sparse patches | high-res input | selector map | sparse patches |

Figure 3: **Looking at Adaptive Computation.** We visualize the selector's prediction of *where* to compute and the extractor's sparse input for *what* to see. Left: Spatially-sparse traffic sign recognition. Right: Fine-grained bird recognition. The same selector generalizes across these images and tasks.

Table 1: **ImageNet Classification.** LookWhere is more accurate and significantly faster than SoTA adaptive computation methods for ViTs. Inference memory is equal for all models. Each cell reports the results for ViT-S/ViT-B: $-$ = not reported and $L$ = layer.

| Method | Top-1 Acc. % ↑ | FLOPs G ↓ | Speed K im/s ↑ | Avg Tok per $L$ ↓ |
|---|---|---|---|---|
| DINOv2 [1] | 81.9/84.2 | 6.2/23.6 | 6.8/2.2 | 256 |
| PiToMe [23] ($r$=0.925) | 79.8/ - | 3.1/ - | 6.6/ - | 128 |
| DTEM [25] ($r$=16) | 79.4/80.7 | 2.4/9.2 | 7.0/3.8 | 94 |
| ATC [24] ($r$=0.9) | 79.9/82.0 | 4.0/15.3 | 0.4/0.4 | 169 |
| LTRP [31] (keep=0.75) | - /82.8 | - /18.3 | - /2.3 | 147 |
| **LookWhere** ($k$=128) | 80.3/83.0 | 3.8/14.8 | 9.5/3.2 | 128 |

Table 2: **ADE20K Segmentation.** We compare three levels of adaptive computation ($r$, $k$): LookWhere is more accurate & efficient at each level.

| Method | mIoU % ↑ | FLOPs G ↓ | Speed K im/s ↑ |
|---|---|---|---|
| DINOv2 [1] | 46.8 | 46.7 | 0.9 |
| DTEM [25] ($r$=0.5) | 38.9 | 19.0 | 0.7 |
| **LookWhere** ($k$=512) | 40.6 | 14.6 | 2.0 |
| DTEM [25] ($r$=0.4) | 42.6 | 22.3 | 0.6 |
| **LookWhere** ($k$=768) | 43.3 | 23.1 | 1.4 |
| DTEM [25] ($r$=0.3) | 44.3 | 25.8 | 0.5 |
| **LookWhere** ($k$=1024) | 44.6 | 32.8 | 1.0 |

## 3 Experiments: Accuracy & Efficiency

We evaluate on standard visual recognition benchmarks in Sec. 3.1, to confirm that LookWhere is generally accurate and efficient, and on high-resolution benchmarks in Sec. 3.2, to measure adaptive computation when efficiency is key. The standard benchmarks, ImageNet-1K classification [32] and ADE20K semantic segmentation [33], are ubiquitous in deep learning for vision. The high-resolution benchmarks, Swedish Traffic Signs [34], Caltech-UCSD Birds [41], and Billiard Balls [29] evaluate spatially-sparse classification tasks as established by existing work on adaptive computation. We then analyze our choice of self-supervision and ablate our selector, extractor, and distillation in Sec. 3.4.

Fig. 3 shows our selector's choice of high-resolution patches (see Appendix A.5 for more examples). Recall that we only update our extractor for each task, and rely on the selector to generalize when predicting where to compute.

**Comparisons and Baselines.** We choose state-of-the-art comparisons encompassing both token *reduction* (PiToMe [23], ATC [24], and DTEM [25]) and token *selection* (DPS [29], and IPS [30]). We review their sophisticated adaptive computation methods in Sec. 4, and Appendix A.1.

**Measuring Computation.** We measure throughput speed, FLOPs, and peak memory usage for a complete accounting of computational efficiency. We profile computation with the official `torch.profiler` tool, the reference implementations of all methods, and an RTX 4090 GPU.

### 3.1 Comparison Experiments on Standard Recognition Benchmarks

We apply our standard finetuning of LookWhere (Sec. 2.2) to update the extractor for image classification on ImageNet-1K [42] and semantic segmentation on ADE20K [33]. We include DINOv2 as a non-adaptive baseline and as the teacher for LookWhere during pretraining.

**ImageNet Classification.** This standard benchmark of accuracy and efficiency is also used for adaptive computation with ViTs (Tab. 1). We follow prior work [43, 25, 23, 44] and finetune for 30 epochs at $224^2$ px resolution. LookWhere achieves the best accuracy and speed for adaptive computation at ViT-S and ViT-B scales with comparable memory and FLOPs. For top speed with the smaller ViT-S model, it delivers $1.36\times$ faster computation than the second fastest method (DTEM).

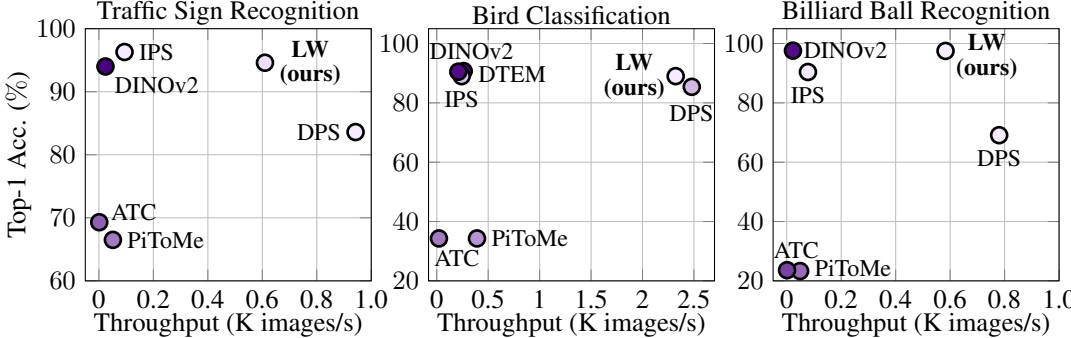

Figure 4: **High-Resolution Recognition on Traffic Signs, Birds (CUB), and Billiards.** We plot the accuracy and throughput of LookWhere, its teacher (DINOv2), and SoTA token reduction and selection methods. LookWhere reaches the Pareto frontier for sparse object recognition (left, right) and for fine-grained classification (center). To more fully measure computation, points are colored by memory usage (darker is higher/less efficient). See Appendices A.2 and A.3 for more details.

**ADE20K Segmentation.** This standard benchmark of semantic segmentation / pixel classification (Tab. 2) requires more spatial precision than image classification. Most adaptive computation methods do not tackle this more challenging task, but DTEM [25] does, so we compare in their setting. We vary the amount of computation to measure the accuracy/efficiency trade-off. LookWhere has better accuracy at $\geq 2\times$ higher speed compared to DTEM.

For both tasks, adaptive computation improves efficiency, but with a trade-off. Without adaptive computation, DINOv2 is the most accurate but less efficient. With adaptive computation, LookWhere achieves the best accuracy and highest efficiency.

## 3.2 High-Resolution Benchmarks with Spatial Sparsity: Where Efficiency is Key

We evaluate the accuracy, efficiency, and generalization of LookWhere on high-resolution inputs. Specifically we choose datasets with spatial sparsity or fine-grained detail, and finetune while limiting the ratio of high-resolution tokens computed. For these experiments we set the number of patches $k$ for LookWhere to process only $10\%$ of the high-resolution input. This sparsity at high-resolution tests our selector's ability to prioritize and improve efficiency by adaptive computation.

In these experiments, we control for factors like backbone architecture (DINOv2), training time (30 epochs of finetuning), and input size (square crops following DINOv2 pretraining). We standardize in this way for clarity of comparison; please see the end of this section for more details.

**Spatially-Sparse Recognition of Traffic Signs.** The Traffic Signs dataset [34] has large images of signs and class labels. The variable and often small size of the signs results in spatial sparsity. We follow the established adaptive computation benchmark [29, 30, 45] of recognizing speed limit signs as 50, 70, or 80 km/h or no sign. We set the input size to be square: $994\times994$ px.

LookWhere strikes the best accuracy-speed balance across all methods (Fig. 4, left). LookWhere is competitive with the state-of-the-art: it rivals IPS in accuracy while reducing inference time by $6\times$ and FLOPs by $34\times$, although IPS reaches $1.1\%$ higher accuracy with more computation. LookWhere exceeds its teacher DINOv2 in efficiency *and* accuracy, even though the teacher processes the full input. Note that longer training and additional tuning can achieve higher accuracy for all methods (including LookWhere, DPS, and IPS) but departs from this controlled setting for fair comparisons.

**Fine-Grained Recognition of Birds.** We evaluate how well LookWhere represents visual detail by experimenting with fine-grained recognition on the Caltech-UCSD Birds (CUB-200-2011) dataset [41] of 200 bird species in 11,788 images. This common benchmark has been adopted for adaptive computation [29, 46]. We set the input size to square $518\times518$ px following the default for DINOv2.

LookWhere reaches the top accuracy, least FLOPs, and fastest speed among adaptive computation methods (Fig. 4, center). Relative to its DINOv2 teacher, which processes all patches, LookWhere has nearly equal accuracy at $\frac{1}{10}$ of the patches for more efficiency in speed, memory, and FLOPs.

**Sparse Recognition and Inter-patch Reasoning with Billiards.** The Billiard Balls dataset [29] is a synthetic benchmark designed to test spatial reasoning. Each image contains 4-8 numbered balls, and

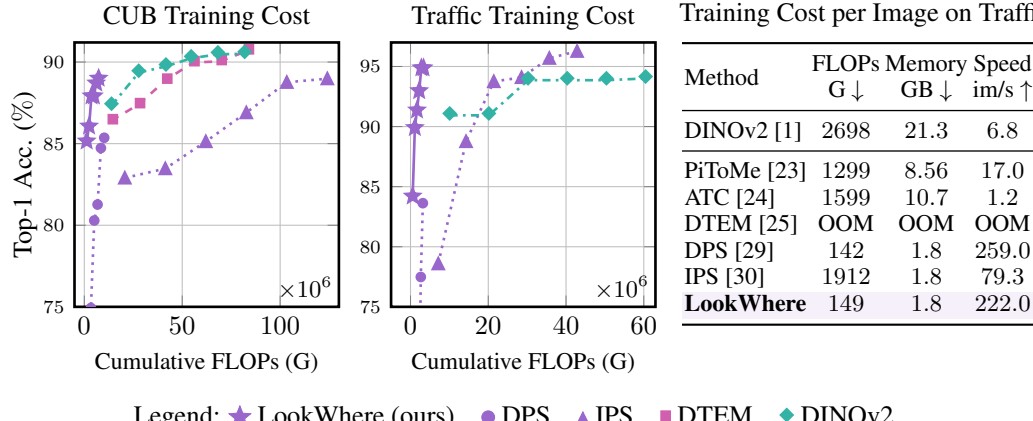

Figure 5: **Finetuning Computation.** We plot (left) the accuracy and total FLOPs used during training for adaptive computation methods on high-res data, and report (right) the cost per image. LookWhere rivals or bests the accuracy of SoTA token reduction and selection methods at a fraction of the cost.

the task is to identify the larger number between the left-most and right-most balls. We set the input size to be square: $1008 \times 1008$ px.

LookWhere delivers the best accuracy-speed trade-off: it nearly maintains its DINOv2 teacher's accuracy with significantly fewer FLOPs, while also rivaling the speed of DPS (Fig. 4, right). DPS achieves higher throughput by incurring a $28\%$ drop in accuracy.

**Experiment Setup for Controlled Comparisons.** We finetune all models for 30 epochs with the AdamW optimizer [47]. All methods use the DINOv2 (ViT-Base) backbone, which is the same as the teacher for LookWhere. We tune all methods equally. For each method, we sweep learning rates $\{2e-5, 5e-5, 8e-5, 1e-4\}$, selecting the best result based on final test accuracy. See Appendix A.2 for hyperparameter settings (e.g., token reduction/selection amounts) and exhaustive results.

## 3.3 Pretraining

We pretrain on ImageNet-1K [42]. We train ViT-Base for 200 epochs and ViT-Small for 400 epochs. We augment the data with standard RandAugment [48] transformations. We evaluate the teacher on every input for its attention and representation. We set the selector's input size $R_{\text{low}}=154$ and depth $L_{\text{low}}=3$ for efficiency; these perform well, and LookWhere works with different choices. At each step, we sample $k \in [16, 128]$ (out of $N_{\text{high}}^2=1{,}369$ tokens per image) for transferrable adaptive computation at high resolution. We optimize by AdamW [47] with the same learning rate for the selector and extractor. Please see Appendix A.2 for full details.

Our what-where distillation is generally robust to pretraining settings. It works well across learning rates, loss weights, selector sizes, and the choice of low-res tokens shared by the selector & extractor.

## 3.4 Analysis and Ablations

**Accounting for Computation.** Efficiency depends on input size and hardware acceleration. For time, LookWhere is faster at low-res (e.g., $224^2$ px) and higher-res (e.g., $>512^2$ px). For FLOPs, LookWhere is often equal or better, but at low-res its selector is relatively costly, and the total can exceed the FLOPs of methods without such a predictor. For memory, LookWhere is equal or better at low-res and high-res, and can reduce training memory by $>5\times$. While these metrics usefully summarize computation, on close examination, the type and hardware compatibility of operations impact efficiency. LookWhere relies only on the accelerated GPU computation of standard ViT layers. However, PiToMe, DTEM, and ATC all rely on clustering algorithms, which are FLOP efficient yet slower on current hardware. As a result, at high-res LookWhere is more time/FLOP/memory efficient than these SoTA methods with $>10\times$ the speed and $>5\times$ reduction in FLOPs and memory (Fig. 5).

**Adversarial Robustness.** To check if the sparse processing due to the selector has an effect on robustness, we evaluate LookWhere and and its teacher against the standard attacks in AutoAttack

[49]. More specifically we evaluate LookWhere and DINOv2 models finetuned on ImageNet-1K against attacks on 10K images from ImageNet-Val. LookWhere achieves robust accuracies of $0.75\%$ and $16.6\%$, while DINOv2 achieves $0.11\%$ and $7.17\%$ using $L_\infty$-bounded attacks with epsilons $\frac{1}{255}$ and $\frac{0.5}{255}$ respectively. Note that we do not claim practical adversarial robustness, and that these are weak attacks relative to the standards for adversarial attack research. No adversarial training was done and these are simply nominal models trained by regular finetuning. Nonetheless, LookWhere is significantly more robust to these attacks at small epsilon than DINOv2, which shows an effect of our selector-extractor architecture and sparse computation. We include this exploratory experiment to encourage further work at the intersection of adaptive computation with adversarial robustness, in case robustness gains could compound.

### 3.4.1 Where to Look for Supervision of Where to Look?

Our selector drives adaptive computation by predicting where to process high-res patches. How the selector learns is fundamental to our method, so we investigate the choice of supervision for it.

**Attention Maps as Oracle Selectors.** ViTs compute *thousands* of 2D attention maps over patch tokens. We test 35 salient choices of teacher attention maps and their aggregations as substitutes for the selector. Specifically, we pretrain LookWhere ViT-S extractors by selecting high-res patches using teacher attention maps instead of selector predictions. These experiments train on ImageNet for 100 epochs. To check generality across tasks, we evaluate via $k$NN for classification over class token representations on ImageNet-HR [50] (a high-res ImageNet-1K test set) and $k$NN for segmentation over patch token representations on ADE20K.

**Results across Layers and Queries.** Fig. 6 shows how accuracy depends on which layers and query tokens to choose for the attention maps. For layers, accuracy generally improves with depth. For query tokens, classification is insensitive ($<\pm1\%$), but segmentation is more sensitive ($>\pm3$). We choose the teacher's last layer attention over patch tokens: it balances the accuracy of both tasks.

| **Classification** | Layer Aggregation | | | | |
|---|---|---|---|---|---|
| Query Aggregation | | first half | last half | last third | last only | all layers |
| | cls | 69.3 | 72.1 | 72.4 | 73.4 | 71.9 |
| | reg | 69.1 | 70.2 | 71.3 | 73.4 | 69.6 |
| | pat | 67.0 | 71.9 | 72.0 | 73.1 | 71.6 |
| | cls+reg | 69.3 | 70.4 | 72.7 | 73.7 | 71.2 |
| | cls+pat | 69.6 | 72.6 | 73.0 | 73.7 | 72.0 |
| | reg+pat | 68.9 | 71.3 | 72.1 | 74.3 | 70.9 |
| | cls+reg+pat | 69.2 | 72.2 | 73.0 | 73.4 | 71.6 |

| **Segmentation** | Layer Aggregation | | | | |
|---|---|---|---|---|---|
| Query Aggregation | | first half | last half | last third | last only | all layers |
| | cls | 64.3 | 63.8 | 63.4 | 63.4 | 64.5 |
| | reg | 64.5 | 64.9 | 65.0 | 64.1 | 65.2 |
| | pat | 65.5 | 67.4 | 67.3 | 66.2 | 67.5 |
| | cls+reg | 64.5 | 64.7 | 64.8 | 63.2 | 65.2 |
| | cls+pat | 65.4 | 65.7 | 64.9 | 63.7 | 65.7 |
| | reg+pat | 65.1 | 66.0 | 66.1 | 64.6 | 66.1 |
| | cls+reg+pat | 64.9 | 65.8 | 65.2 | 63.8 | 65.8 |

Figure 6: **Teacher Attention for Selection.** We evaluate teacher attention maps as selector maps to choose distillation targets. For classification ($k$NN on ImageNet-HR), the deepest attention maps and class queries perform best. For segmentation ($k$NN on ADE20K), averaging over all layers and patch queries perform best. We choose the last layer's patch token attention maps to balance the tasks.

### 3.4.2 Ablating Selector-Extractor Architecture and What-Where Distillation

**Design Choices.** We ablate four design choices of our architecture and pretraining (Fig. 7). We experiment with $518^2$ resolution, ViT-S architecture, and $k=72$ (out of $1,369$ high-res patches). (1) We vary the depth and input size of the selector (a). Larger inputs tend to help more than deeper architecture, so we choose $154^2$ px resolution a 3 layers. (2) We experiment with training the selector map end-to-end from the extractor's distillation losses alongside the attention distillation loss. We try REINFORCE [52] (inspired by PatchDrop [28]), and Gumbel Top-$k$ [51] for this purpose and observe no improvement from either (b). (3) We confirm the effectiveness of passing global tokens from the selector to the extractor for efficiency (c). (4) We measure the trade-off in class token vs. patch token loss weights across classification and segmentation (see Appendix A.4).

**Random Selection Ablation.** We ablate our selector by selecting high-res patches at random. On Traffic Signs, accuracy **drops by 25**%, which is only marginally better than choosing the majority class. The drop on CUB is $11\%$, and on ImageNet, it is $2.1\%$ (Fig. 7(d)). The smaller drop on ImageNet could be due to its object-centric images making *where* less important.

(a) Deeper selectors & larger inputs are more accurate but slower.

| depth | res. | cls. | seg. |
|---|---|---|---|
| 1 | $252^2$ | 66.2 | 60.5 |
| 3 | $252^2$ | 69.7 | 64.7 |
| 3 | $154^2$ | 68.8 | 62.7 |
| 6 | $154^2$ | 69.7 | 63.0 |
| 6 | $98^2$ | 63.0 | 60.2 |
| 12 | $98^2$ | 69.6 | 61.0 |
| 12 | $42^2$ | 47.3 | 54.6 |

(b) Training the selector map on teacher attention *alone* is best.

| case | cls. | seg. |
|---|---|---|
| none (stop grad) | 63.0 | 60.2 |
| Gumbel Top-K [51] | 61.4 | 58.9 |
| REINFORCE [52] | 61.8 | 59.6 |

(c) Giving the extractor global tokens from the selector helps.

| case | cls. | seg. |
|---|---|---|
| none | 60.7 | 56.9 |
| cls-only | 62.2 | 58.7 |
| reg-only | 62.0 | 59.8 |
| cls+reg | 63.0 | 60.2 |

(d) Random selection hurts. Varying $k$ during finetuning enables flexible inference at different $k$.

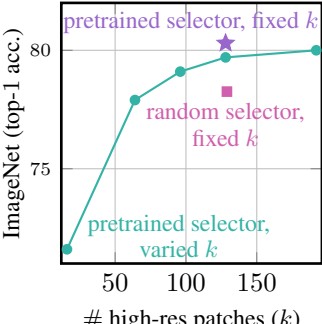

Figure 7: **Ablating Architecture and Training** "cls." = $k$NN image/*class token* classification on ImageNet-HR (top-1 acc.) and "seg." = $k$NN local/*patch token* classification on ADE20K (top-1 acc.). Default settings are shared across ablations; final setting is used on downstream experiments. Subfigure (d) finetunes on ImageNet and evaluates on ImageNet-Val for comparability with Tab. 1.

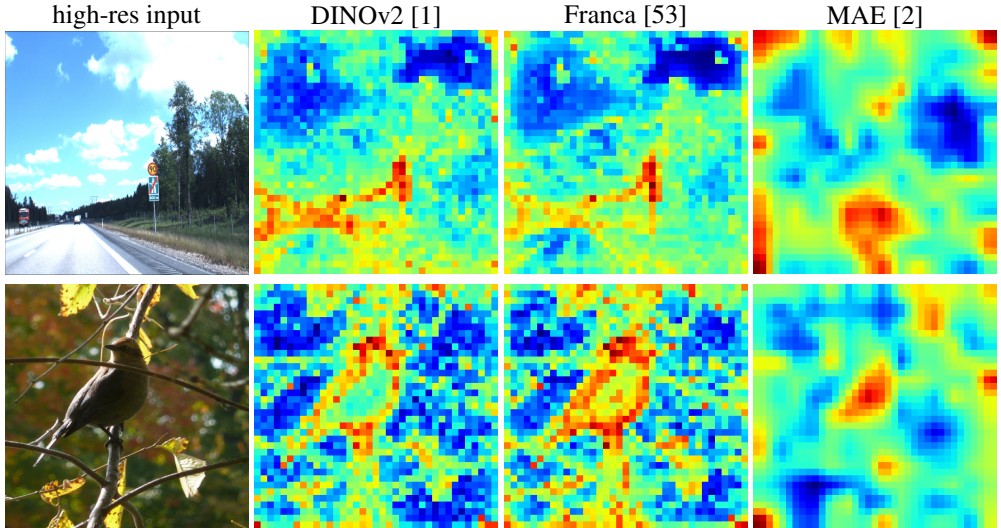

Figure 8: **Alternative Teachers.** We visualize attention maps from different teachers for *where* to look. DINOv2 and Franca both identify the visually interesting and serve as effective teachers for LookWhere. By contrast, MAE's attention is occasionally unfocused, making it a weaker teacher.

**Varying $k$ for Flexibility.** We randomly vary $k$ by sampling from $[1,256]$ during finetuning on ImageNet. This enables choosing $k$ at inference for faster/more accurate computation (Fig. 7(d)).

**Pretraining Using Other Teachers.** Thus far our teacher has been DINOv2. However, attending to the visually interesting is not specific to DINOv2: as Figure 8 demonstrates, other models may also guide *where* to look (see Appendix A.6 for more examples). Here, we experiment with a Franca [53] teacher and pretrain a selector-extractor using our what-where distillation. Specifically, the selector's target is Franca's last layer CLS-token attention (chosen because the Franca paper shows it is meaningful). The extractor's target is Franca's last layer representation. We evaluate LookWhere-Franca's extracted features with $k$NN on ImageNet-HR: it achieves $57.0\%$ / $68.9\%$ / $71.1\%$ vs. LookWhere-DINOv2's $73.1\%$ / $80.3\%$ / $81.8\%$ for $k$ high-res patches 16 / 72 / 128 (both sets of results are at the ViT-Base scale). LookWhere-Franca thus performs well enough as a feature extractor but worse than LookWhere-DINOv2. We also finetune LookWhere-Franca on the Traffic Signs, Birds, and Billiard Balls datasets and report results in Appendix A.2. In summary, LookWhere-Franca outperforms prior work but not LookWhere-DINOv2. To make LookWhere fully

up-to-date, we train a LookWhere-DINOv3 [54] to achieve $43.8\%$ / $73.4\%$ / $77.6\%$ accuracy. Better results using Franca or DINOv3 teachers can surely be achieved by searching for better selector targets, e.g. our DINOv2 search (Fig. 6), which we leave for future work.

## 4 Related Work

**Token** *reduction* drops or merges already-computed tokens layer-by-layer to save *further* computation. Many of these methods cleverly combine similar tokens by soft bipartite matching algorithms [22, 23, 26, 55, 25], while others incorporate dynamic filtering [56, 57] or differentiable pruning [44]. Nonetheless, such *gradual* reduction requires processing all tokens at the first layer, nearly all at the second, and so on. This is already too much in our case: *any* processing of *all* tokens is too expensive at high resolution. By contrast, our selector processes a low-resolution input, and then our extractor receives a small set of high-resolution patches. LookWhere never processes the full-resolution input to attain efficiency unobtainable by reduction.

**Token** *selection* chooses the inputs for computation, and so can save more, but must learn how to select and do so efficiently. Prior works scale to megapixel images but require complex and expensive per-task optimization [28, 29, 30]. IPS [30] iterates over sets of patches to identify the most salient, which reduces memory but still takes time. DPS [29] and PatchDrop [28] train predictors that process low-resolution versions of the input to score high-resolution patches by discrete optimization. While efficient for inference, their optimization requires sampling, careful tuning of gradient approximation [52, 58], and multi-stage training (PatchDrop). LookWhere also processes low and high resolutions for efficient inference, but with simple and efficient training during both pretraining (by distillation) and finetuning (by extractor-only updates).

**Self-supervision for Adaptive Computation.** Most existing token reduction and selection methods rely on task-specific optimization that does not harness transfer across tasks. LTRP [31] instead pretrains a general selector by self-supervision, like LookWhere, but lacks a general extractor and scores patches differently. Specifically, LTRP relies on multiple forward passes over perturbed inputs to measure the patch-wise sensitivity of a self-supervised model (MAE [2]) to supervise its selector. This results in expensive pretraining ($>10\times$ ours) without an extractor to transfer. By contrast, LookWhere directly distills the teacher's attention and representation, which only needs *one* forward pass to compute. Furthermore, LookWhere simultaneously pretrains the selector and extractor for efficient re-use and transfer across tasks.

**Distillation.** Many works train the attention maps of students to mimic the attention maps of teachers [59, 60, 61, 62], and last layer teacher attention maps in particular [63]. However, our what-where distillation trains our student, the selector, to predict a single map that summarizes multiple teacher attention maps. This strategy allows our selector to predict a map at higher resolution than its input— e.g., receiving $N_{\text{low}} \times N_{\text{low}}$ patches and predicting an $N_{\text{high}} \times N_{\text{high}}$ map. Furthermore, existing methods do not distill attention for adaptive computation: we are the first to turn self-supervised attention maps into the supervision of how to prioritize tokens for adaptive computation.

## 5 Conclusion

We introduce our selector-extractor architecture and what-where distillation for accurate, efficient, and general adaptive computation on diverse visual recognition tasks. LookWhere is simple but effective: a selector predicts a 2D map (*where* to look) for an extractor that predicts full-res image representations (*what* to see). This simplicity enables efficiency on current hardware: LookWhere applies standard transformer operations used just right. Together, our selector & extractor represent images they never fully see for efficient finetuning and deployment, especially at high resolutions.

**Limitations and Future Work.** There is more to learn for selection and there are more dimensions for adaptive computation. LookWhere does not finetune the selector. Although this achieves SoTA results, by pretraining a task-general selector, finetuning a task-specific selector could do better. Our selector-extractor is only spatial, but could be temporal (LookWhen) for video or spectral (LookWhich) for remote sensing bands.

## Acknowledgements

AF is primarily supported by an NSERC PGS-D scholarship. ES is supported by a Canada CIFAR AI Chair. JW acknowledges the support of NSERC, RGPIN-2024-05357. YY is primarily supported by an Ontario Graduate Scholarship (OGS) and a Vector Scholarship in AI. Resources used in preparing this research were provided, in part, by the Province of Ontario, the Government of Canada through CIFAR, and companies sponsoring the Vector Institute. We thank the Google TPU Research Cloud (TRC) for providing the TPUs on which we conducted all pretraining experiments. We thank Eric Tzeng for their helpful review and feedback on the experiments and analysis.

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

# A  Appendix

## A.1  Adaptive Token Method Descriptions

### A.1.1  Token Reduction

**Protect Informative Tokens before Merging (PiToMe)** [23] builds on Bipartite Soft Matching (BSM), which partitions tokens into two sets and merges pairs across both according to key-based similarity. Instead of considering *all* tokens for merging, PiToMe proposes an energy score—defined as an aggregate similarity of a token with all others—to distinguish redundant (high-energy) from informative (low-energy) tokens. Only the top $2k$ high-energy tokens are considered for merging via BSM, preserving the most informative tokens.

**Agglomerative Token Clustering (ATC)** [24] replaces BSM with a bottom-up, hierarchical clustering approach for token merging. Each token starts as its own cluster, and clusters are iteratively merged based on a similarity score defined between clusters. At each step, the two most similar (i.e., redundant) clusters are merged, continuing until a target number of tokens remains.

**Decoupled Token Embedding for Merging (DTEM)** [25] introduces a trainable module that decouples token merging from representation learning. It projects intermediate token embeddings to extract merging-specific features for estimating pairwise token similarities. Unlike traditional *hard* assignment grouping schemes, DTEM uses a differentiable top-$k$ operator [64] to softly group tokens, followed by BSM for merging. Notably, the soft merging retains *all* tokens during training, with discretization applied only during inference for actual token reduction.

### A.1.2  Patch Selection

**PatchDrop** [28] formulates selection as a Reinforcement Learning problem to train a patch selection policy. Given a down-sampled input, the policy returns Bernoulli distributions over high-resolution patches from which samples are drawn. The policy is initially trained using the REINFORCE [52] algorithm with a *fixed* pretrained classifier, receiving rewards that encourage both classification accuracy and sparse sampling. A joint finetuning stage then updates the policy and classifier together for improved performance.

**Differentiable Patch Selection (DPS)** [29] casts patch selection as a ranking problem, where a scoring network assigns relevance scores to high-resolution patches based on a low-resolution input. A differentiable top-$k$ module, based on the perturbed maximum method [58], selects the top $k$ scoring patches for downstream processing by feature and patch-aggregation networks. The entire pipeline is trained end-to-end under supervision.

**Iterative Patch Selection (IPS)** [30] performs ranking-based patch selection directly on the high-res input through sequential glimpses. A scoring module maintains a buffer of the top $k$ most salient patches, which is updated auto-regressively by evaluating $I$ patches at a time in no-gradient mode. The selected patches are then re-embedded with gradients enabled and passed to a cross-attention transformer-based classifier, enabling end-to-end training. This same cross-attention module is shared with the scoring network to guide saliency prediction.

**Learning to Rank Patches (LTRP)** [31] trains a patch ranking model via self-supervision. Using a pretrained masked autoencoder, the method randomly masks patches and estimates each *visible* patch's importance by measuring the change in image reconstruction when the patch is removed—yielding a pseudo-relevance score. A ranking model is then trained to predict the resulting ranks such that, following pretraining, they may be used to retain the top most salient patches for downstream tasks.

## A.2  Experimental Details

### A.2.1  Pretraining Hyperparameters

We pretrain our ViT-S for 400 epochs and ViT-B for 200 epochs on ImageNet-1K [32] using the AdamW optimizer [47]. We use a batch size of 1024, image size of $518{\times}518$ px, weight decay of 0.05, learning rate warmup of 10%, peak learning rate of $2e{-}4$ with cosine decay to $1e{-}05$, and RandAugment [48] data augmentation.

### A.2.2 ImageNet Hyperparameters

We finetune for 30 epochs on ImageNet-1K [32] using the AdamW optimizer [47]. We use a batch size of 128, image size of 224×224 px, weight decay of 0.02, learning rate warmup of 10%, peak learning rate of 2e−5 with cosine decay to 1e−5, and 3-Augment [65] data augmentation.

### A.2.3 ADE20K Hyperparameters

We finetune for 160K steps on ADE20K [33] using the AdamW optimizer [47]. We use a batch size of 16, image size of 518×518 px, weight decay of 0.02, learning rate warmup of 10%, peak learning rate of 2e−5 with cosine decay to 1e−5, and data augmentation: random crop scale [0.7, 1.0], horizontal flip, random choice {gray scale, solarize, gaussian blur}, and color jitter (0.3). We compute all patch-token representations by interpolating sparse representations with the same interpolation method used during pretraining. We use a simple linear head that maps patch-token representations to pixel-wise class predictions.

### A.2.4 Traffic Signs Recognition

**Traffic Signs Setup.** We use the annotated subset of the Swedish Traffic Signs dataset [34], containing 747 training and 684 test images. All images are resized and cropped to square dimensions of 994×994 px for compatibility with DINOv2 ViT backbones pretrained on square images. IPS and DPS use 980×980 px due to compatibility with their ViT patch extraction methods. We finetune all models for 30 epochs using the AdamW optimizer [47]. We sweep learning rates over {2e−5, 5e−5, 8e−5, 1e−4} with a batch size of 16, following [30], and report the best final test accuracy achieved by each method. We apply a weight decay of 0.02, a 10% learning rate warmup, with cosine decay to 1e−05, and data augmentation including random cropping (scale range [0.8, 1.0]) and standard RandAugment transformations [48]. DTEM and PiToMe merge tokens after every layer, ATC merges after layers 4, 7, and 10 (following [56]).

**Tabular Traffic Signs Results.** Table 3 reports full experimental results to complement Figure 4.

Table 3: **Traffic Signs results.** LookWhere remains competitive with SoTA token selection methods, being more efficient than IPS and rivaling DPS for speed, memory, and FLOPs. LookWhere-R uses random masking instead of the selector, but still receives the selector's low-res global tokens for context. DTEM runs out of memory (OOM).

| Method | Top-1 Acc. | Memory (GB) ↓ | | FLOPs (G) ↓ | | Speed (im/s) ↑ | |
|---|---|---|---|---|---|---|---|
| | % ↑ | Test | Train | Test | Train | Test | Train |
| DINOv2 [1] | 94.0 | 3.1 | 21.3 | 900 | 2698 | 24.1 | 6.8 |
| PiToMe [23] (r=0.9) | 66.5 | 3.1 | 8.6 | 440 | 1300 | 51.4 | 17.0 |
| ATC [24] (r=0.7) | 69.3 | 2.7 | 10.7 | 537 | 1599 | 1.3 | 1.2 |
| DTEM [25] | OOM | OOM | OOM | OOM | OOM | OOM | OOM |
| DPS (r=0.1) [29] | 83.6 | 0.8 | 1.8 | 49 | 142 | 942.4 | 259.0 |
| IPS (r=0.1) [30] | 96.3 | 0.8 | 1.8 | 1819 | 1912 | 94.1 | 79.3 |
| **LookWhere-R** (k=504) | 70.6 | 0.8 | 1.8 | 53 | 149 | 609.8 | 222.0 |
| **LookWhere** (k=504) | 94.6/90.2 | 0.8 | 1.8 | 53 | 149 | 609.8 | 222.0 |
| **LookWhere** (k=1008) | 95.2/91.4 | 0.9 | 2.5 | 110 | 320 | 288.1 | 99.1 |

*LookWhere accuracies are reported for DINOv2/Franca teachers. Compute is the same.

### A.2.5 Fine-grained Bird Classification

**Fine-grained Bird Classification Setup.** We finetune for 30 epochs on the Caltech-UCSD Birds (CUB-200-2011) dataset [41] of 200 bird species in 11,788 images. We sweep learning rates over {2e−5, 5e−5, 8e−5, 1e−4} using a batch size of 64. Images are resized to 518×518 px by default (588×588 px for IPS and DPS for 98×98 pixel patches fed to the ViT feature extractor). We use a weight decay of 0.02, a 10% learning rate warmup, and cosine decay to a minimum learning rate of 1e−05. Data augmentation includes random cropping (scale range [0.8, 1.0]) and standard RandAugment transformations [48]. DTEM and PiToMe merge tokens after every layer, ATC merges after layers 4, 7, and 10 (following [56]).

**Tabular Bird Results.** Table 4 reports full experimental results to complement Figure 4.

Table 4: **Bird results.** LookWhere achieves the Pareto frontier with the fastest speed, and lowest memory and FLOPs among SoTA token reduction and selection methods while retaining competitive accuracy. LookWhere-R uses random masking instead of the selector, but still receives the selector's low-res global tokens for context.

| Method | Top-1 Acc. | Memory (GB) ↓ | | FLOPs (G) ↓ | | Speed (im/s) ↑ | |
|---|---|---|---|---|---|---|---|
| | % ↑ | Eval | Train | Eval | Train | Eval | Train |
| DINOv2 [1] | 90.5 | 0.9 | 3.1 | 152 | 455 | 209.4 | 67.4 |
| PiToMe [23] $(r=0.9)$ | 34.3 | 0.9 | 1.9 | 81 | 239 | 391.1 | 128.4 |
| ATC [24] $(r=0.7)$ | 34.4 | 0.9 | 2.1 | 95 | 285 | 21.2 | 18.2 |
| DTEM [25] $(r=96)$ | 90.7 | 1.0 | 5.5 | 84 | 464 | 261.8 | 52.3 |
| DPS $(r=0.1)$ [29] | 85.4 | 0.7 | 1.8 | 19 | 57 | 2479.0 | 697.7 |
| DPS $(r=0.2)$ | 88.3 | 0.7 | 1.8 | 33 | 98 | 1442.2 | 432.1 |
| IPS $(r=0.1)$ [30] | 89.0 | 0.8 | 1.8 | 653 | 690 | 261.8 | 213.8 |
| IPS $(r=0.2)$ | 90.2 | 0.8 | 1.8 | 639 | 704 | 241.8 | 179.4 |
| **LookWhere-R** $(k=136)$ | 78.2 | 0.8 | 1.7 | 16 | 41 | 2322.4 | 865.6 |
| **LookWhere** $(k=136)$ | 89.0/86.6 | 0.8 | 1.7 | 16 | 41 | 2322.4 | 865.6 |
| **LookWhere** $(k=273)$† | 90.1/88.1 | 0.8 | 1.7 | 29 | 79 | 1333.6 | 479.0 |
| **LookWhere** $(k=684)$† | 90.4/88.2 | 0.8 | 1.9 | 71 | 206 | 515.8 | 173.6 |

*LookWhere accuracies are reported for DINOv2/Franca teachers. Compute is the same.
†Pretrained with a wider sampled patch range (256-512) to better support larger K at finetuning.

### A.2.6 Billiard Ball Inter-patch Reasoning

**Billiard Ball Setup.** We finetune for 30 epochs on the Billiard Ball dataset [29] of 8k training images, and 10k for test. Each image contains 4-8 balls, numbered 1 through 9, with the task of identifying the larger number among the left-most and right-most balls. We resize images to $1008 \times 1008$ px for compatibility with DINO's patch size; DPS and IPS use $1072 \times 1072$ px for compatibility with their ViT patch extraction methods. We train most methods with a batch size of 64 following [29], and 32 for IPS / DPS due to memory constraints. The remaining setup follows identically from Traffic Signs (see A.2.4).

**Tabular Billiard Ball Results.** Table 5 reports full experimental results to complement Figure 4.

Table 5: **Billiard Ball results.** LookWhere attains SoTA accuracy on the Billiard Balls task, matching DINO's performance while receiving $\frac{1}{10}$ of the patches to significantly improve efficiency; achieving the least memory and FLOP consumption and speeds comparable to DPS. LookWhere-R uses random masking instead of the selector, but still receives the selector's low-res global tokens for context.

| Method | Top-1 Acc. | Memory (GB) ↓ | | FLOPs (G) ↓ | | Speed (im/s) ↑ | |
|---|---|---|---|---|---|---|---|
| | % ↑ | Test | Train | Test | Train | Test | Train |
| DINOv2 [1] | 97.6 | 2.9 | OOM | 939 | OOM | 22.9 | 6.4 |
| PiToMe [23] $(r=0.7)$ | 23.3 | 3.2 | 8.9 | 458 | 1353 | 49.1 | 15.8 |
| ATC [24] $(r=0.7)$ | 23.6 | 3.2 | 11.2 | 559 | 1667 | 1.2 | 1.0 |
| DTEM [25] | OOM | OOM | OOM | OOM | OOM | OOM | OOM |
| DPS $(r=0.1)$ [29] | 69.1 | 0.8 | 1.8 | 59 | 170 | 780.0 | 215.5 |
| IPS $(r=0.1)$ [30] | 90.4 | 0.8 | 1.8 | 1810 | 1921 | 77.9 | 65.2 |
| **LookWhere-R** $(k=518)$ | 27.8 | 0.8 | 1.8 | 55 | 154 | 583.1 | 209.6 |
| **LookWhere** $(k=518)$ | 97.3/97.1 | 0.8 | 1.8 | 55 | 154 | 583.1 | 209.6 |
| **LookWhere** $(k=1037)$ | 97.5/97.0 | 0.9 | 2.5 | 114 | 331 | 285.4 | 98.5 |
| **LookWhere** $(k=2592)$ | 97.4/97.0 | 1.4 | 7.0 | 350 | 1041 | 77.0 | 25.8 |

*LookWhere accuracies are reported for DINOv2/Franca teachers. Compute is the same.

## A.3   Additional Finetuning Experiments

We further assess the generalization of LookWhere's selector by evaluating on more diverse, medical and remote sensing datasets.

### A.3.1   Remote Sensing Classification

**AID Setup.** We finetune for 30 epochs on the Aerial Image Dataset (AID) [66] of 10k images for aerial scene classification across 30 classes. We use a 50/50 train/test split, batch size of 64, and resize images from $600{\times}600$ px to $602{\times}602$ px to match DINOv2's patch size. The remaining setup follows identically from Traffic Signs (see A.2.4). Results are reported in Table 6.

Table 6: **AID results.** LookWhere rivals DINOv2's performance on remote sensing classification using a fraction of the patches. Accuracy improves with increased computation.

| Method | Top-1 Accuracy (%) ↑ |
|---|---|
| DINOv2 [1] | 98.7 |
| **LookWhere** $(k=185)$ | 97.4 |
| **LookWhere** $(k=370)$ | 98.0 |
| **LookWhere** $(k=925)$ | 98.3 |

### A.3.2   Medical Chest X-ray Classification

**NIH Chest X-ray Setup.** We use the NIH Chest X-ray [67] dataset of 112k medical images (90k train / 22k test) for multi-label clinical diagnosis across 14 labels. Images are resized from $1024{\times}1024$ px to $1022{\times}1022$ px and finetuning for 10 epochs using a batch size of 64. The remaining setup follows identically from Traffic Signs (see A.2.4). Following prior work [68, 69], we report AUC-ROC scores in Table 7.

Table 7: **NIH Chest X-ray results.** On multi-label medical classification, LookWhere remains competitive with DINOv2 despite processing only a fraction of the patches. Performance scales with the allocated computation.

| Method | AUC-ROC ↑ |
|---|---|
| DINOv2 [1] | 82.2 |
| **LookWhere** $(k=533)$ | 78.7 |
| **LookWhere** $(k=1066)$ | 79.6 |
| **LookWhere** $(k=2665)$ | 81.1 |

### A.4   Detailed Ablation Setup and Results

We ablate several design choices, including: (1) teacher attention targets for selector maps, (2) distillation loss weights, (3) selector map training schemes, (4) low-resolution conditioning tokens for the extractor, and (5) the input size and depth of the selector. Unless stated otherwise, all ablations are conducted over $400$ epochs of pretraining on ImageNet-1K [32] at a resolution of $518{\times}518$ pixels using the ViT-S architecture. During pretraining, we sample $k \in [16, 128]$ uniformly at random, and report results at test time for fixed values $k \in \{16, 72, 128\}$ for general classification and segmentation downstream tasks. Unless stated otherwise, ablations use our default settings, which are highlighted in blue.

To ablate LookWhere efficiently, we use a simple $k$NN-based evaluation. For classification, we process all ImageNet-HR [50] images with LookWhere and store the resulting class-token representations $\hat{z}^{cls}_{\text{high}}$. We then compute top-1 accuracy using a leave-one-out $k$NN approach; specifically, for each ImageNet-HR image, we retrieve the 3 nearest neighbors, excluding the query itself and assign the most frequent class among them as the prediction. For segmentation, we process all ADE20K [33] images with LookWhere and store 10 patch-token representations $\hat{z}^{pat}_{\text{high}}$ per sample. We then compute top-1 accuracy using the validation set as queries and the training set as keys. We fetch 20 patch tokens for each query and assign the most common class as the predicted label.

### A.4.1 Teacher Attention Targets for Selection

To distill the teacher's attention, we extract the unnormalized attention among its patch tokens at *select* layers and query tokens and then average over layers, queries and heads (Fig. 9). This approximates where the selected patches contributed to the teacher representation. In this section, we explore different combinations of *layers* and query *tokens* used for attention aggregation.

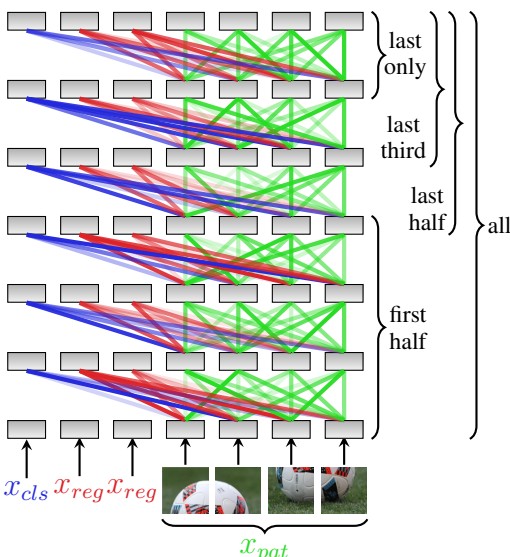

Figure 9: **Attention Aggregation.** We aggregate attention by averaging unnormalized attention scores originating from the `cls`, `reg`, and `patch` queries across different layers (all, first 3, last 3, last 2, and last only). Score maps from each query *type* are weighted equally when several are used (i.e., we average the maps among tokens of each type first, before averaging across types).

**Classification** Layer Aggregation

| Query Aggregation | first half | last half | last third | last only | all layers |
|---|---|---|---|---|---|
| cls | 48.5 | 52.9 | 54.8 | 57.6 | 52.1 |
| reg | 47.8 | 50.1 | 50.8 | 56.5 | 49.2 |
| pat | 46.9 | 49.8 | 50.3 | 54.1 | 48.8 |
| cls+reg | 48.5 | 49.0 | 53.8 | 57.2 | 50.9 |
| cls+pat | 48.9 | 52.2 | 54.2 | 57.7 | 50.8 |
| reg+pat | 47.5 | 49.4 | 51.0 | 56.3 | 48.6 |
| cls+reg+pat | 47.9 | 51.5 | 52.8 | 57.3 | 51.3 |

**Segmentation** Layer Aggregation

| Query Aggregation | first half | last half | last third | last only | all layers |
|---|---|---|---|---|---|
| cls | 52.3 | 51.8 | 51.6 | 51.8 | 52.1 |
| reg | 53.4 | 53.2 | 52.7 | 52.2 | 53.7 |
| pat | 53.6 | 54.2 | 53.6 | 53.3 | 54.1 |
| cls+reg | 53.4 | 52.2 | 52.5 | 51.7 | 53.0 |
| cls+pat | 52.5 | 52.7 | 52.6 | 52.3 | 52.7 |
| reg+pat | 53.6 | 53.1 | 53.6 | 53.2 | 53.9 |
| cls+reg+pat | 53.8 | 53.1 | 52.1 | 52.0 | 53.4 |

Figure 10: **Teacher Attention for Selection** ($k=16$). We evaluate teacher attention maps as selector maps to choose distillation targets when testing using $k=16$ selected patches.

**Classification** — Layer Aggregation (Query Aggregation)

| | first half | last half | last third | last only | all layers |
|---|---|---|---|---|---|
| cls | 65.0 | 68.6 | 69.2 | 70.3 | 67.8 |
| reg | 64.1 | 66.2 | 66.8 | 70.2 | 65.5 |
| pat | 62.4 | 69.0 | 68.9 | 70.4 | 67.5 |
| cls+reg | 65.4 | 66.4 | 68.8 | 70.4 | 67.1 |
| cls+pat | 65.0 | 68.9 | 69.6 | 70.9 | 68.3 |
| reg+pat | 64.1 | 67.5 | 68.4 | 70.6 | 66.6 |
| cls+reg+pat | 64.9 | 68.0 | 69.2 | 70.6 | 67.6 |

**Segmentation** — Layer Aggregation (Query Aggregation)

| | first half | last half | last third | last only | all layers |
|---|---|---|---|---|---|
| cls | 61.4 | 60.8 | 60.4 | 59.4 | 60.8 |
| reg | 61.1 | 61.5 | 61.6 | 60.5 | 62.1 |
| pat | 62.1 | 64.2 | 63.7 | 62.7 | 64.2 |
| cls+reg | 61.2 | 61.0 | 61.0 | 60.1 | 62.2 |
| cls+pat | 62.0 | 62.0 | 61.0 | 60.9 | 62.0 |
| reg+pat | 61.7 | 62.4 | 62.7 | 61.1 | 62.6 |
| cls+reg+pat | 61.5 | 62.2 | 61.4 | 60.5 | 62.4 |

Figure 11: **Teacher Attention for Selection** ($k=72$). We evaluate teacher attention maps as selector maps to choose distillation targets when testing using $k=72$ selected patches.

**Classification** — Layer Aggregation (Query Aggregation)

| | first half | last half | last third | last only | all layers |
|---|---|---|---|---|---|
| cls | 69.3 | 72.1 | 72.4 | 73.4 | 71.9 |
| reg | 69.1 | 70.2 | 71.3 | 73.4 | 69.6 |
| pat | 67.0 | 71.9 | 72.0 | 73.1 | 71.6 |
| cls+reg | 69.3 | 70.4 | 72.7 | 73.7 | 71.2 |
| cls+pat | 69.6 | 72.6 | 73.0 | 73.7 | 72.0 |
| reg+pat | 68.9 | 71.3 | 72.1 | 74.3 | 70.9 |
| cls+reg+pat | 69.2 | 72.2 | 73.0 | 73.4 | 71.6 |

**Segmentation** — Layer Aggregation (Query Aggregation)

| | first half | last half | last third | last only | all layers |
|---|---|---|---|---|---|
| cls | 64.3 | 63.8 | 63.4 | 63.4 | 64.5 |
| reg | 64.5 | 64.9 | 65.0 | 64.1 | 65.2 |
| pat | 65.5 | 67.4 | 67.3 | 66.2 | 67.5 |
| cls+reg | 64.5 | 64.7 | 64.8 | 63.2 | 65.2 |
| cls+pat | 65.4 | 65.7 | 64.9 | 63.8 | 65.7 |
| reg+pat | 65.1 | 66.0 | 66.1 | 64.6 | 66.1 |
| cls+reg+pat | 64.9 | 65.8 | 65.2 | 63.8 | 65.8 |

Figure 12: **Teacher Attention for Selection** ($k=128$). We evaluate teacher attention maps as selector maps to choose distillation targets when testing using $k=128$ selected patches.

### A.4.2 Classification Accuracy vs. Distillation Losses

We ablate distillation weights and feature-interpolation parameters with 100 epochs of pretraining. During pretraining (and when using ADE20K downstream), we interpolate the sparse/visible high-res patch-token representations to compute a full 2D grid. We compute the predicted dense patch-token representations as the weighted sum of the $N$ nearest neighbors in 2D space, among the sparse/visible patch tokens. The weights in the weighted sum are the Euclidean distances raised to the $\text{pow}^{th}$ power, then normalized s.t. all weights sum to 1. We experiment with $N \in \{5, 16\}$ and $\text{pow} \in \{1, 2\}$, and summarize our results in Tables 8 through 11.

Table 8: **Distillation Losses** ($\text{pow}=1$, 5 neighbors). We evaluate different combinations of distillation loss weights for pretraining the selector-extractor ($\lambda_{cls}, \lambda_{pat}$) and selector map ($\lambda_{map}$). We consider spatial patch interpolation using Euclidean distance, with 5 selected neighbours.

| $\lambda_{cls}$ | $\lambda_{pat}$ | $\lambda_{map}$ | $\mathcal{L}_{map}$ | cls. (by $k$) | | | seg. (by $k$) | | |
|---|---|---|---|---|---|---|---|---|---|
| | | | | 16 | 72 | 128 | 16 | 72 | 128 |
| 1 | 1 | 1 | KL | 47.7 | 61.7 | 66.4 | 51.6 | 59.6 | 62.6 |
| 1 | 1 | 0.1 | KL | 46.8 | 61.5 | 66.7 | 51.0 | 59.5 | 62.3 |
| 1 | 0.1 | 1 | KL | 48.1 | 62.5 | 67.4 | 48.2 | 57.3 | 60.8 |
| 1 | 0.1 | 0.1 | KL | 48.9 | 63.0 | 67.4 | 47.8 | 56.7 | 60.7 |
| 0.1 | 1 | 1 | KL | 44.9 | 60.3 | 66.1 | 51.6 | 59.8 | 63.4 |
| 0.1 | 1 | 0.1 | KL | 45.0 | 60.2 | 65.3 | 51.3 | 59.7 | 62.9 |
| 0.1 | 0.1 | 1 | KL | 46.1 | 62.2 | 66.9 | 51.0 | 59.5 | 63.1 |
| 0.1 | 0.1 | 0.1 | KL | 48.7 | 62.0 | 66.7 | 50.8 | 59.8 | 62.5 |
| 1 | 1 | 1 | MSE | 50.2 | 61.8 | 66.2 | 51.2 | 58.5 | 62.0 |

Table 9: **Distillation Losses** ($\text{pow}=1$, 16 neighbors). We evaluate different combinations of distillation loss weights for pretraining the selector-extractor ($\lambda_{cls}, \lambda_{pat}$) and selector map ($\lambda_{map}$). We consider spatial patch interpolation using Euclidean distance, with 16 selected neighbours.

| $\lambda_{cls}$ | $\lambda_{pat}$ | $\lambda_{map}$ | $\mathcal{L}_{map}$ | cls. (by $k$) | | | seg. (by $k$) | | |
|---|---|---|---|---|---|---|---|---|---|
| | | | | 16 | 72 | 128 | 16 | 72 | 128 |
| 1 | 1 | 1 | KL | 48.9 | 62.2 | 66.8 | 49.4 | 58.1 | 61.5 |
| 1 | 1 | 0.1 | KL | 48.1 | 61.9 | 66.9 | 48.9 | 57.4 | 61.0 |
| 1 | 0.1 | 1 | KL | 47.4 | 62.4 | 67.0 | 47.4 | 56.7 | 60.2 |
| 1 | 0.1 | 0.1 | KL | 48.3 | 62.2 | 66.2 | 47.5 | 55.9 | 60.2 |
| 0.1 | 1 | 1 | KL | 44.6 | 60.4 | 65.5 | 49.8 | 58.8 | 62.0 |
| 0.1 | 1 | 0.1 | KL | 45.8 | 60.0 | 66.0 | 49.2 | 58.5 | 61.6 |
| 0.1 | 0.1 | 1 | KL | 46.5 | 62.6 | 67.0 | 49.0 | 58.2 | 62.1 |
| 0.1 | 0.1 | 0.1 | KL | 47.8 | 62.5 | 67.1 | 49.6 | 58.4 | 61.6 |

Table 10: **Distillation Losses** ($\text{pow}=2$, 5 neighbors). We evaluate different combinations of distillation loss weights for pretraining the selector-extractor ($\lambda_{cls}, \lambda_{pat}$) and selector map ($\lambda_{map}$). We consider spatial patch interpolation using squared Euclidean distance, with 5 selected neighbours.

| $\lambda_{cls}$ | $\lambda_{pat}$ | $\lambda_{map}$ | $\mathcal{L}_{map}$ | cls. (by $k$) | | | seg. (by $k$) | | |
|---|---|---|---|---|---|---|---|---|---|
| | | | | 16 | 72 | 128 | 16 | 72 | 128 |
| 1 | 1 | 1 | KL | 46.7 | 61.9 | 67.2 | 50.7 | 59.0 | 62.8 |
| 1 | 1 | 0.1 | KL | 47.0 | 61.5 | 65.8 | 50.5 | 59.6 | 62.5 |
| 1 | 0.1 | 1 | KL | 49.3 | 62.6 | 67.0 | 48.3 | 56.7 | 60.6 |
| 1 | 0.1 | 0.1 | KL | 48.0 | 62.6 | 67.2 | 48.3 | 56.7 | 60.6 |
| 0.1 | 1 | 1 | KL | 44.8 | 60.6 | 66.0 | 48.0 | 56.5 | 48.0 |
| 0.1 | 1 | 0.1 | KL | 43.9 | 60.3 | 65.1 | 51.3 | 59.8 | 63.2 |
| 0.1 | 0.1 | 1 | KL | 45.7 | 62.3 | 67.0 | 50.6 | 59.0 | 62.5 |
| 0.1 | 0.1 | 0.1 | KL | 47.6 | 62.3 | 66.3 | 50.7 | 59.1 | 62.6 |

Table 11: **Distillation Losses** (`pow=2`, 16 neighbors). We evaluate different combinations of distillation loss weights for pretraining the selector-extractor ($\lambda_{cls}, \lambda_{pat}$) and selector map ($\lambda_{map}$). We consider spatial patch interpolation using squared Euclidean distance, with 16 selected neighbours.

| $\lambda_{cls}$ | $\lambda_{pat}$ | $\lambda_{map}$ | $\mathcal{L}_{map}$ | cls. (by $k$) | | | seg. (by $k$) | | |
|---|---|---|---|---|---|---|---|---|---|
| | | | | 16 | 72 | 128 | 16 | 72 | 128 |
| 1 | 1 | 1 | KL | 48.3 | 62.1 | 66.9 | 50.2 | 58.2 | 61.7 |
| 1 | 1 | 0.1 | KL | 47.6 | 61.5 | 66.4 | 49.8 | 57.5 | 61.6 |
| 1 | 0.1 | 1 | KL | 48.7 | 62.6 | 67.5 | 47.7 | 56.7 | 60.1 |
| 1 | 0.1 | 0.1 | KL | 49.1 | 62.6 | 66.9 | 47.6 | 6.1 | 59.8 |
| 0.1 | 1 | 1 | KL | 44.1 | 61.0 | 65.3 | 50.4 | 59.5 | 62.6 |
| 0.1 | 1 | 0.1 | KL | 44.2 | 60.8 | 65.7 | 50.4 | 58.8 | 61.7 |
| 0.1 | 0.1 | 1 | KL | 45.9 | 62.3 | 67.2 | 49.8 | 58.4 | 62.2 |
| 0.1 | 0.1 | 0.1 | KL | 47.8 | 62.5 | 67.1 | 50.3 | 58.7 | 62.2 |

### A.4.3   Selector Map Training

In addition to distilling the teacher's attention map, we explore differentiable learning of the selector map to better leverage the extractor's signal for optimal patch selection. To enable differentiable selection, we experiment with (1) Gumbel Top-K [51] and (2) REINFORCE [52].

In both approaches, the selector map is treated as logits defining a softmax probability distribution over patches; we sample the map using the Gumbel Top-K trick. We make sampling differentiable through the straight-through estimator (i.e., differentiable Gumbel Top-K) and sweep over learning rates and sampling temperatures. With REINFORCE, we model pretraining as a one-step (contextual) bandit problem: low-resolution images define the state, actions are binary masks selecting $k$ patches (as in PatchDrop [28]), and the reward is derived from the loss. Specifically, we compute an advantage as the difference in distillation loss between a greedy top-$k$ policy and the sampled action. We backpropogate through the extractor for both samples. We explore various learning rates, weightings for distillation and policy gradient losses, and the use of an entropy bonus to promote exploration.

We observe that training fails to converge without a map distillation loss (which effectively acts analogous to a KL regularizing term with respect to the teacher's "policy"). Table 12 reports the best results across all configurations, consistently showing that teacher distillation alone performs best. Gumbel Top-K pretrains for 400 epochs, while REINFORCE pretrains for 300 epochs (to roughly balance compute across all methods since REINFORCE backpropogates through both the policy sample and greedy baseline).

Table 12: **Selector Map Training Schemes**. We consider different training schemes for the selector map *in addition to distillation*. Stop grad refers to no additional scheme, while Gumbel Top-K and REINFORCE leverage sampling to estimate a gradient from the extractor's signal. We find that distillation, alone, is optimal.

| case | cls. (by $k$) | | | seg. (by $k$) | | |
|---|---|---|---|---|---|---|
| | 16 | 72 | 128 | 16 | 72 | 128 |
| none (stop grad) | 50.9 | 63.0 | 66.5 | 51.1 | 60.2 | 62.9 |
| Gumbel Top-K [51] | 44.5 | 61.4 | 66.2 | 49.1 | 58.9 | 62.5 |
| REINFORCE [52] | 47.9 | 61.8 | 66.0 | 50.7 | 59.6 | 62.8 |

### A.4.4   Selector Layer Initialization

We explored several selector initialization schemes as alternatives to truncating the teacher's first $L_{low}$ layers. The results, summarized in Table 13, show that LookWhere is generally robust to initialization, especially at larger K values. At smaller K values, we find performance improves slightly when initializing with deeper layers.

Table 13: **Selector Initialization**. We experiment with different layer initialization schemes for LookWhere's selector. We find that performance is relatively unchanged for larger K, while smaller K benefits slightly from initializing with deeper layers.

| case | cls. (by $k$) | | | | seg. (by $k$) | | | |
|---|---|---|---|---|---|---|---|---|
| | 16 | 72 | 128 | 256 | 16 | 72 | 128 | 256 |
| 0-1-2 | 38.52 | 55.90 | 62.76 | 68.68 | 48.10 | 57.88 | 60.33 | 63.81 |
| 3-4-5 | 39.88 | 56.68 | 62.20 | 68.36 | 47.78 | 57.58 | 60.45 | 64.31 |
| 6-7-8 | 39.90 | 56.66 | 62.42 | 68.44 | 48.06 | 57.77 | 60.24 | 64.24 |
| 9-10-11 | 40.54 | 57.32 | 63.02 | 68.34 | 48.49 | 57.68 | 60.61 | 64.32 |

### A.4.5 Alternative Selector Architectures

LookWhere generalizes to architectures beyond ViTs, although the extractor and selector require different considerations. Here, we demonstrate this flexibility by using a CNN for the selector architecture.

The extractor relies on a partitioning of the input, typically into tokens, that enables: i) selection and processing of sparse inputs, and ii) distillation using partial inputs to learn semantic representations. Input patchification satisfies these needs, making the extractor compatible with various downstream architectures. For instance, methods like DPS and IPS also use patch-based inputs with CNN components.

The selector has a simpler role: approximating the teacher's saliency map. We hypothesize that any modern image processing architecture can perform this task effectively. To validate this, we pretrained two models for 100 epochs: one with our default DINOv2-initialized ViT selector and another with a CNN selector initialized from EfficientNetV2 [70]. As shown in Table 14, the kNN performance is comparable. After finetuning on ImageNet, the EfficientNet-based selector achieved a top-1 accuracy of 82.3%. Overall, these results confirm that LookWhere generalizes well beyond ViT-only designs.

Table 14: **Alternative CNN-based Selectors**. We experiment with using CNN-based Selectors, initialized with EfficientNetV2, for LookWhere. Results are comparable to using a DINOv2-based selector.

| case | cls. (by $k$) | | | seg. (by $k$) | | |
|---|---|---|---|---|---|---|
| | 16 | 72 | 128 | 16 | 72 | 128 |
| DINOv2 [1] | 66.2 | 74.9 | 78.4 | 52.3 | 61.8 | 64.6 |
| EfficientNetV2 [70] | 71.8 | 81.2 | 82.5 | 49.3 | 58.8 | 62.3 |

### A.4.6 Extractor Conditioning on Low-Resolution Global Tokens

To provide the extractor with additional *global* image context, we experiment with initializing its class token $x_{cls}$ and/or register tokens $x_{reg}$ by the selector's final representations for each: $z_{low}^{cls}$ and $z_{low}^{reg}$, respectively. Table 15 summarizes our results; we find that adding global tokens generally helps, although performance interestingly degrades for classification when selecting $k{=}128$ patches specifically.

Table 15: **Extractor Conditioning**. We experiment with feeding the extractor different combinations of global low-resolution tokens resulting from the selector's processing. We find that giving *both* the cls token and all register generally helps performance.

| case | cls. (by $k$) | | | seg. (by $k$) | | |
|---|---|---|---|---|---|---|
| | 16 | 72 | 128 | 16 | 72 | 128 |
| none | 28.2 | 60.7 | 68.4 | 39.7 | 56.9 | 61.1 |
| cls-only | 48.8 | 62.2 | 67.2 | 49.8 | 58.7 | 61.8 |
| reg-only | 49.4 | 62.0 | 67.3 | 51.0 | 59.8 | 62.4 |
| cls+reg | 50.9 | 63.0 | 66.5 | 51.1 | 60.2 | 62.9 |

### A.4.7 Selector Input Size and Depth

We experiment with varying the selector's size by trading off network depth against input resolution while keeping computational cost relatively low. Our results are summarized in Table 16.

Table 16: **Selector Sizes**. Both shallow networks with large inputs and deeper networks with smaller inputs result in suboptimal performance. The best results are achieved by balancing depth and input resolution.

| depth | res. | cls. (by $k$) | | | seg. (by $k$) | | |
|---|---|---|---|---|---|---|---|
| | | 16 | 72 | 128 | 16 | 72 | 128 |
| 1 | $252^2$ | 43.7 | 66.2 | 70.8 | 49.0 | 60.5 | 64.4 |
| 3 | $252^2$ | 65.8 | 72.1 | 73.7 | 56.1 | 64.7 | 67.2 |
| 3 | $154^2$ | 58.7 | 68.8 | 71.2 | 53.9 | 62.7 | 65.1 |
| 6 | $154^2$ | 65.0 | 69.7 | 72.1 | 54.4 | 63.0 | 65.4 |
| 6 | $98^2$ | 50.9 | 63.0 | 66.5 | 51.1 | 60.2 | 62.9 |
| 12 | $98^2$ | 65.8 | 69.6 | 70.9 | 53.0 | 61.0 | 63.3 |
| 12 | $42^2$ | 25.5 | 47.3 | 57.1 | 43.8 | 54.6 | 58.1 |

### A.5 Selector Maps

Our selector generalizes to diverse downstream images and tasks. In particular, we visualize general examples for the recognition of birds (Fig. 13), traffic signs (Fig. 14), and billiard balls (Fig. 15).

### A.5.1 Bird Recognition (CUB)

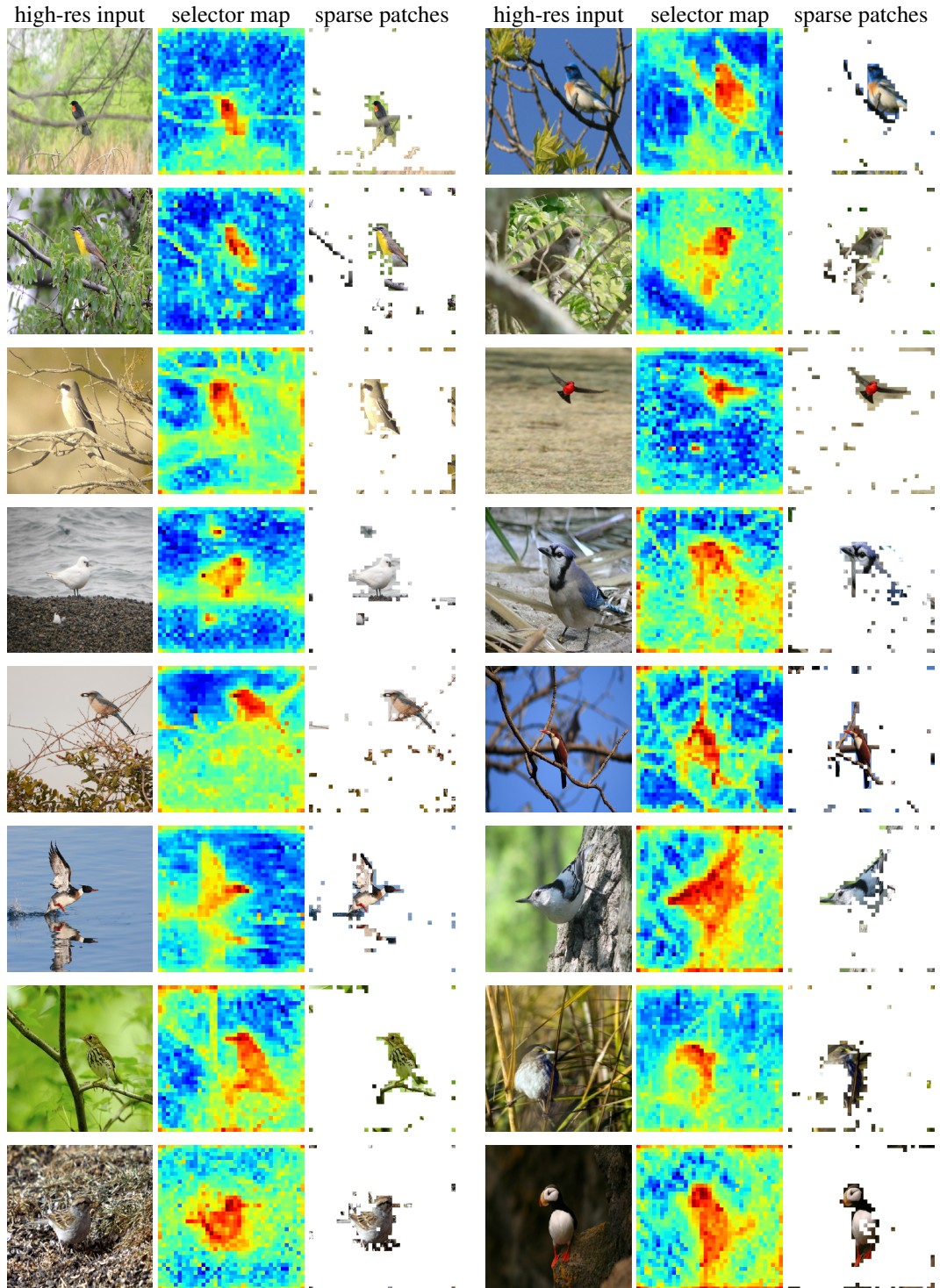

high-res input    selector map    sparse patches      high-res input    selector map    sparse patches

Figure 13: **Adaptive Computation for Recognizing Birds.** We visualize the selector's prediction of *where* to compute and the extractor's sparse input for *what* to see. We do *not* finetune the selector; each pair shows different generalization scenarios, specifically for bird recognition on CUB [41].

## A.5.2 Traffic Signs

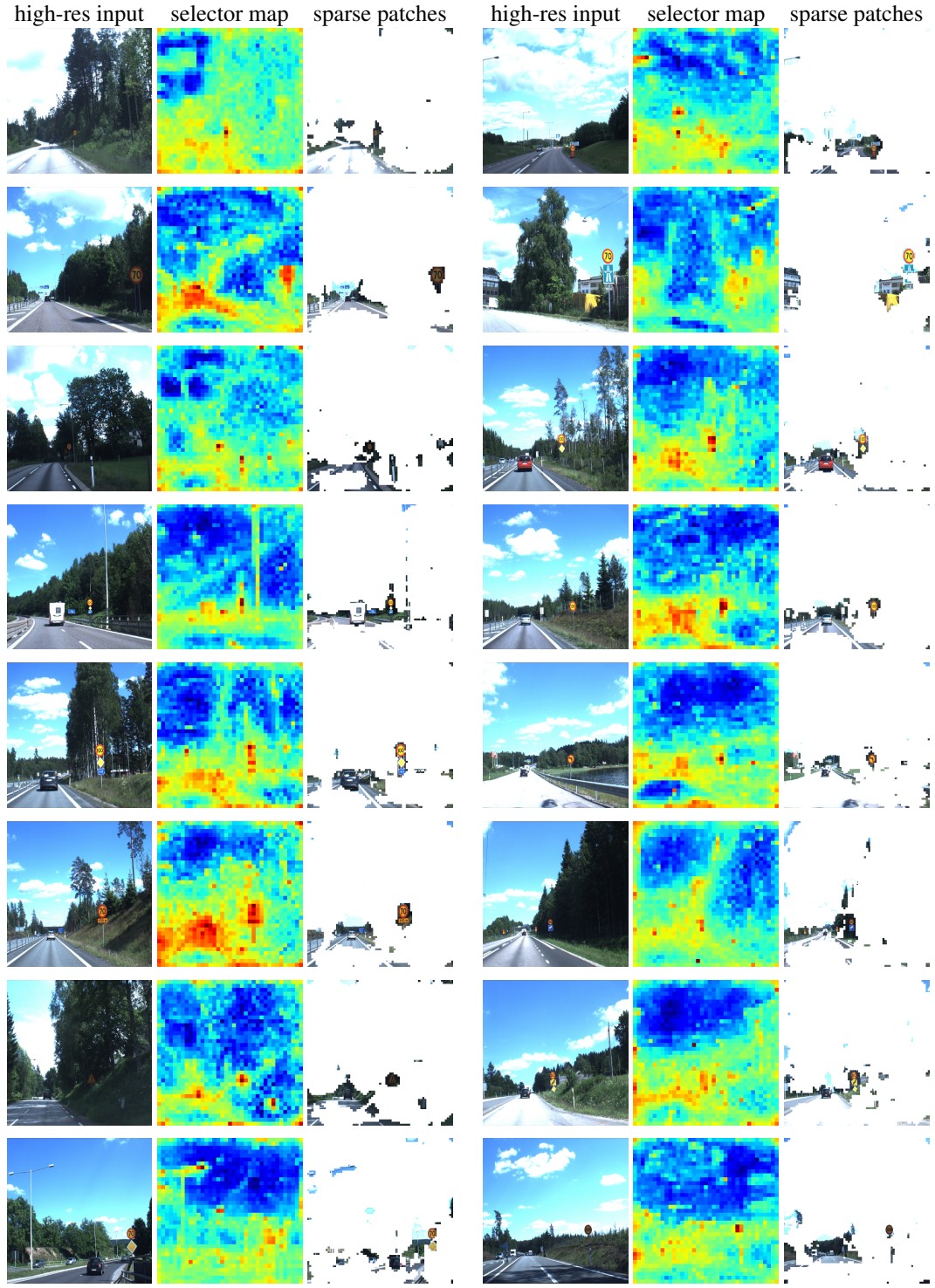

Figure 14: **Adaptive Computation for Traffic Sign Recognition.** We visualize the selector's prediction of *where* to compute and the extractor's sparse input for *what* to see. We do *not* finetune the selector; each pair shows different generalization scenarios, specifically for recognizing traffic signs on the Swedish Traffic Signs dataset [34].

### A.5.3 Billiard Balls

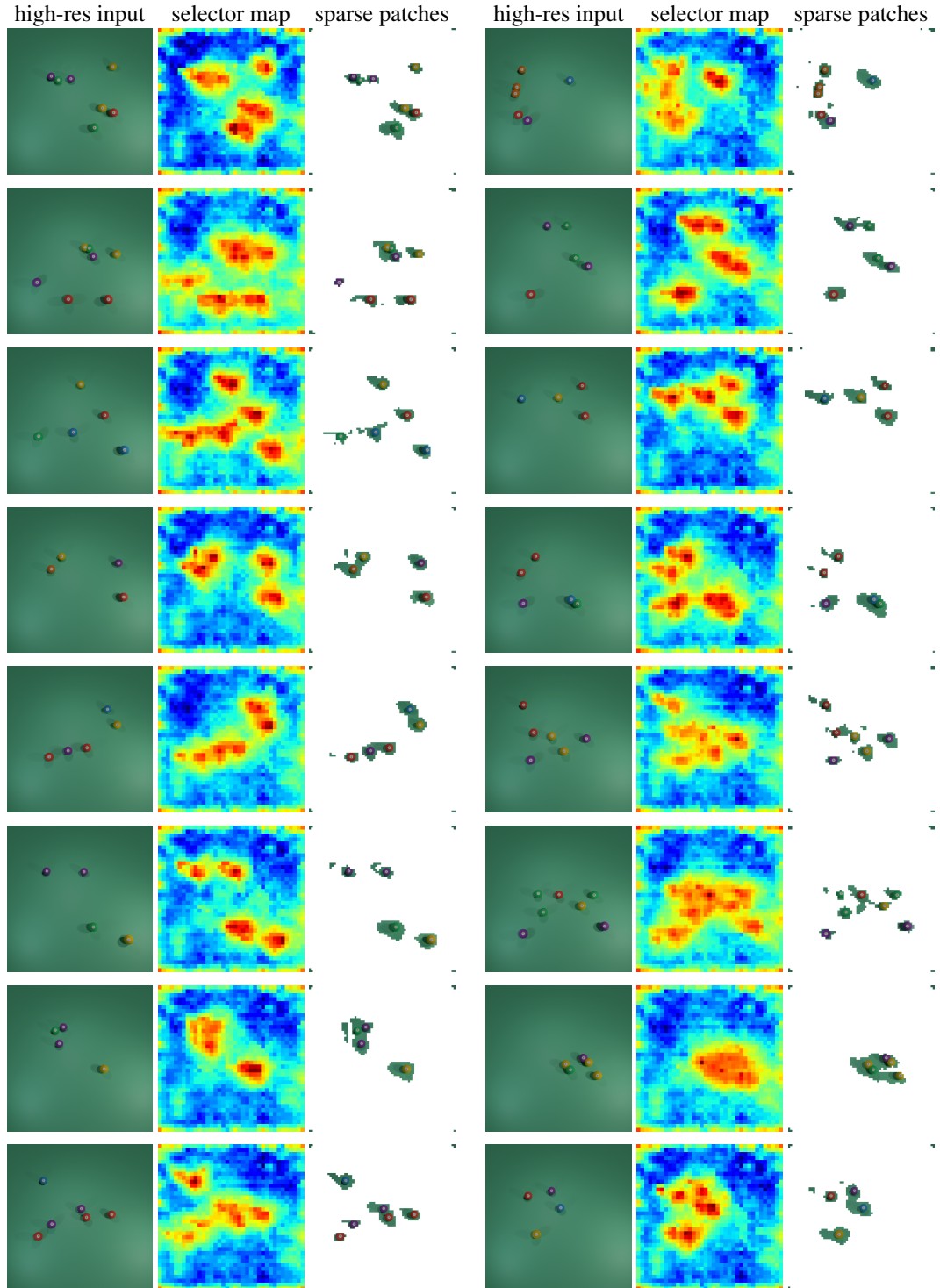

Figure 15: **Adaptive Computation for Billiard Ball Recognition.** We visualize the selector's prediction of *where* to compute and the extractor's sparse input for *what* to see. We do *not* finetune the selector; each pair shows different generalization scenarios, specifically for recognizing and reasoning across several billiard ball configurations [29].

## A.6 Comparing Selector Maps from Different Teachers

LookWhere generalizes to other teachers beyond DINOv2. In particular, we visualize general examples for the recognition of birds (Fig. 16), traffic signs (Fig. 17), and billiard balls (Fig. 18) for three different teachers: DINOv2 [1], Franca [53], and MAE [2] (we upsample MAE's maps to match the resolution of DINOv2/Franca for visual consistency).

### A.6.1 Bird Recognition (CUB)

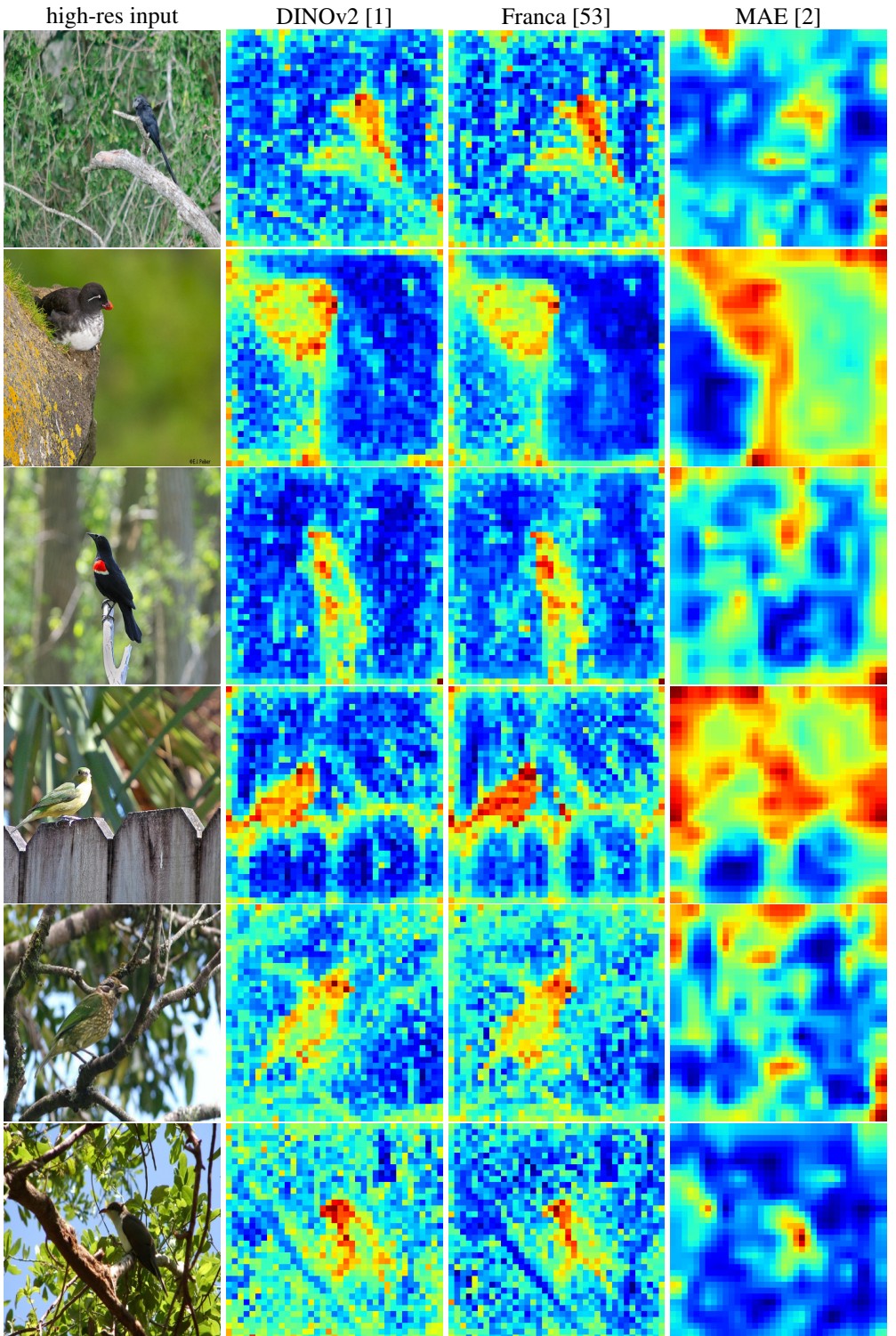

Figure 16: **Alternative Teachers for Recognizing Birds.** We visualize attention maps from different teachers for *where* to look on Bird Recognition tasks.

### A.6.2 Traffic Signs

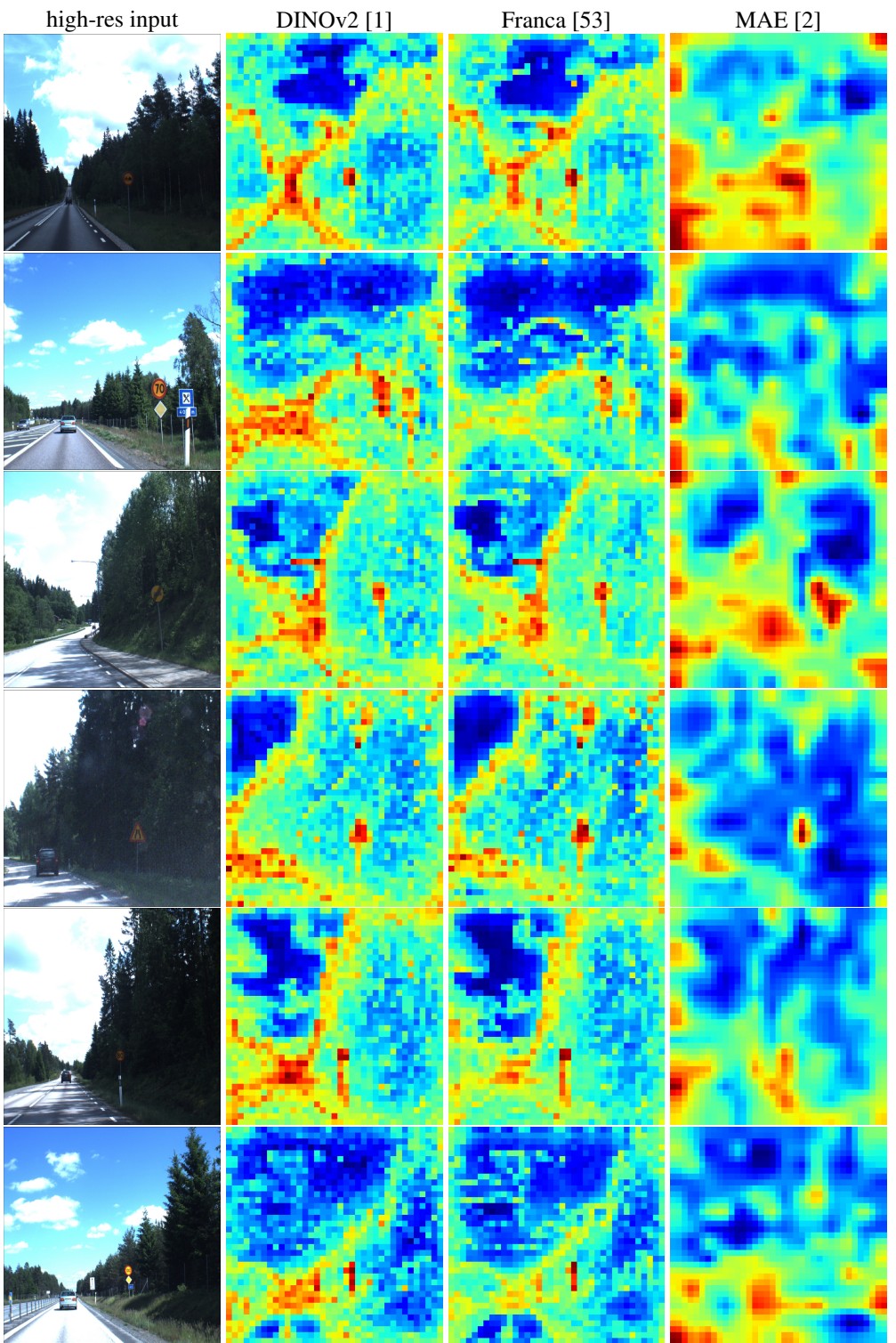

Figure 17: **Alternative Teachers for Traffic Signs.** We visualize attention maps from different teachers for *where* to look on Traffic Signs Recognition.

### A.6.3 Billiard Balls

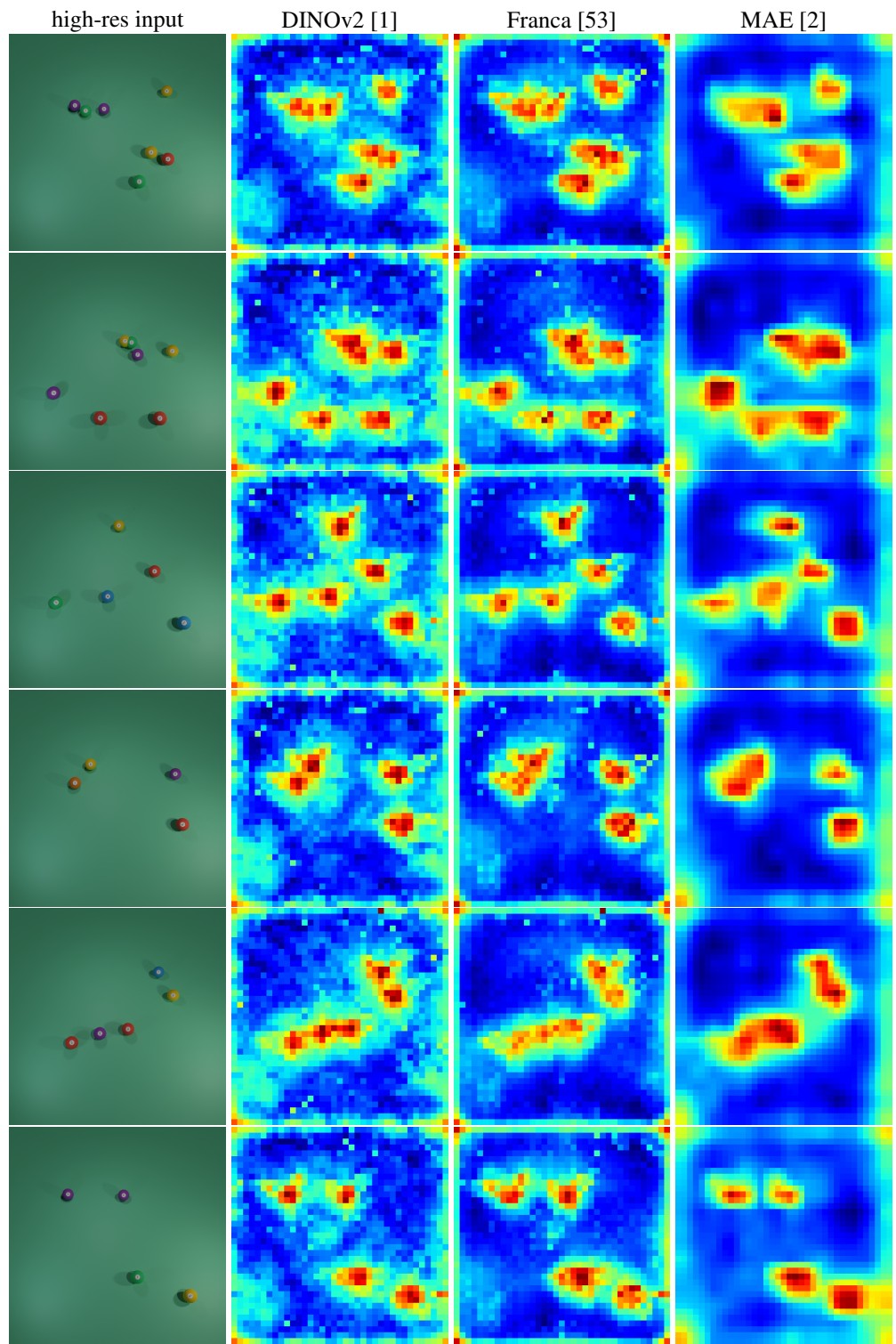

Figure 18: **Alternative Teachers for Billiard Ball Recognition.** We visualize attention maps from different teachers for *where* to look for recognizing and reasoning across several billiard ball configurations [29].

