# OpenReview forum: "LookWhere? Efficient Visual Recognition by Learning Where to Look and What to See from Self-Supervision"
_NeurIPS.cc/2025/Conference — NeurIPS 2025 poster_

### Official Review · Reviewer_7H5B · 2025-06-22

**Clarity:** 2
**Significance:** 3
**Originality:** 3
**Rating:** 4
**Confidence:** 4

**Summary:**

This paper proposes "LookWhere," an adaptive computation method to address the prohibitive computational cost of Vision Transformers, especially on high-resolution images. The method divides computation into a low-cost 'selector' that predicts 'where to look' from a low-resolution input, and an 'extractor' that processes 'what to see' using only the selected high-resolution patches. For training, it employs a novel 'what-where distillation' technique that simultaneously distills the attention (where) and representations (what) from a self-supervised model, enabling task-agnostic generalization and efficient fine-tuning.
Experiments demonstrate that the method achieves a superior accuracy-efficiency trade-off compared to existing methods across diverse recognition benchmarks. Notably, on the high-resolution traffic sign recognition task, it shows remarkable results, reducing FLOPs by up to 34x and inference time by 6x while maintaining accuracy.

**Questions:**

- How much information is lost in the global tokens extracted from a low-resolution input compared to their original high-resolution counterparts? Can it be expected that this loss is small enough to avoid significant performance degradation on tasks that rely heavily on global object information?
- Is the choice of DINOv2 as the teacher model critical? To what extent does the proposed distillation framework depend on the specific properties of DINOv2?

**Ethical Concerns:**

["NO or VERY MINOR ethics concerns only"]

**Limitations:**

yes

**Quality:**

3

**Strengths And Weaknesses:**

**Strengths**

The idea of separating the process into an inexpensive selector that decides "where to look" and an expensive extractor that processes "what to see" is intuitive and compelling. By not needing to process the entire input image at high resolution from the start, it can achieve fundamentally higher computational efficiency than existing token pruning methods.

**Major Weaknesses**

The core of the proposed method lies in reusing global tokens (CLS/register) extracted from a computationally cheap, low-resolution input. While Figure 7(c) demonstrates the effectiveness of this approach, a crucial analysis is absent: a direct comparison with the original tokens from a high-resolution input to quantify how much global contextual information is retained in this low-resolution version.
To convincingly justify the validity of the efficiency gains, it is essential to quantitatively verify that this cost reduction does not incur a significant loss of important contextual information.

**Minor Weaknesses**

- The paper selects DINOv2 as the teacher model. A discussion on why DINOv2 is particularly suitable for this task, and the potential implications of using other self-supervised models, would strengthen the paper's claims of generality. Without this context, it is difficult for the reader to gauge to what extent the framework's effectiveness is tied to the specific properties of DINOv2, as opposed to being a more universally applicable distillation strategy.

- The number of selected patches, k, is a critical hyperparameter that determines the trade-off between accuracy and efficiency. While the paper evaluates several fixed values for k, it does not deeply discuss a mechanism for dynamically determining k based on the task or input image. Although a training method that allows for flexible selection of k at inference is mentioned, the criterion for this selection is left to the user.

---

> ### Author Rebuttal · Authors · 2025-07-31
>
> # Response to Reviewer 7H5B
>
> We thank the reviewer for their detailed and thoughtful comments, and appreciate their recognition of LookWhere’s intuitive design and its ability to achieve strong performance while maintaining computational efficiency. The reviewer raises three key concerns, which we address below.
>
> \
> **How much information is lost in the global tokens extracted from a low-resolution input compared to their high-resolution counterparts?:** \
> This is a valid concern that we are happy to address. First, we clarify that LookWhere not only reuses global tokens from the low-resolution input but also relies heavily on its selector to determine where to look – i.e., to identify the most relevant high-resolution patches. We believe this selective mechanism is also central to LookWhere’s effectiveness.
>
> As emphasized in the introduction, many vision tasks involve spatial redundancy, where only small portions of the image are essential for accurate prediction. LookWhere exploits this sparsity by selecting only those high-resolution patches that receive high attention from the teacher. Low-attention patches contribute less to the global context and can be safely discarded with minimal impact on task performance.
>
> We completely agree with the reviewer that a direct comparison with the original tokens from a high-resolution input is crucial to demonstrate the effectiveness of our approach. However, we believe this is already addressed through our experiments, which all include DINOv2 as a baseline. DINOv2 receives the full high-resolution input, and since LookWhere distills a DINOv2 teacher, the performance of the latter serves as a natural upper bound for LookWhere. As shown in Tables 1 and 2, LookWhere achieves comparable performance to DINOv2 on ImageNet classification and ADE20k segmentation using only a fraction of the patches, supporting that LookWhere’s carefully designed cost reduction does not incur a significant loss of meaningful context. Given this helpful feedback we will revise our explanation of the teacher baselines to clarify that they offer the full high-resolution comparison without sparse processing.
>
> Furthermore, in some downstream tasks, LookWhere even outperforms DINOv2. For example, in Traffic Sign Recognition (Table 3), it achieves higher accuracy with 20% of the patches, and, in the Billiard Ball task (see our response to reviewer J83A), it nearly matches DINOv2’s performance with only 20% of the patches. We attribute this strong performance to the selector’s ability to focus precisely on task-relevant regions.
>
> LookWhere also maintains competitive performance across other downstream tasks – including fine-grained bird classification (Table 4), aerial scene classification, and clinical diagnosis (see new experiments in our response to reviewer J83A). These results further support that our selection strategy retains the _relevant_ global context, even without full high-resolution input.
>
> \
> **To what extent does the proposed distillation framework depend on the specific properties of DINOv2?:** \
> LookWhere relies on DINOv2 – and any other teacher model – in two key ways: to distill where to look via its saliency maps, and what to see via its representations. While many modern models provide high-quality patch representations suitable for training the extractor, our choice of DINOv2 is primarily motivated by its highly semantic and clean attention maps, which allow the selector to identify visually salient regions without supervision from labels or tasks.
>
> Importantly, any teacher model with semantic attention maps could be used effectively within our framework. To test this generality, we conducted new experiments during the rebuttal period using Franca [R1] – a new visual foundation model – with a ViT-B backbone as the teacher, keeping all other hyperparameters fixed. We note that Franca was released after our submission, and demonstrates that LookWhere can be easily applied to and benefit from future models. Franca uses a CLS token for global context and does not employ registers, yet we observed that distillation remains effective. Following Section 3.1, our LookWhere-Franca model with k = 128 (9.3% of tokens) achieves 71.2% accuracy on ImageNet-HR and 61.4% on ADE20K. Furthermore, our finetuned LookWhere-Franca model achieves 81.0% on ImageNet which compares favorably to DINOv2 (Table 1). We also finetuned this LookWhere-Franca model on downstream tasks, obtaining the following results:
> - **Traffic:** 90.2%, 91.4% and 93.4% using K values of 518 (10%), 1036 (20%), and 2592 (50%).
> - **Birds:** 86.6%, 88.1% and 88.2% using K values of 136 (10%), 273 (20%) and 684 (50%).
> - **Billiards:** 97.1%, 97.0%, and 97.0% using K values of 518 (10%), 1036 (20%), and 2592 (50%) (see “Generalization to More Domains” in our response to reviewer J83A)
>
> These results compare favorably to those obtained with DINOv2, and we will reference them alongside our other finetuning results in section 3.2 while also including them near Tables 3 and 4. We agree with the reviewer that discussing the specific advantages of DINOv2, and the general properties favorable within a teacher would strengthen the paper’s generality, and we will incorporate this discussion into Section 2.3 on Pretraining.
>
> \
> **A mechanism for dynamically determining K:** \
> First, we note that – consistent with prior work on token selection using fixed K values – we observe that most instances within a given task can be handled effectively with a similar number of patches. That said, we agree with the reviewer that dynamically selecting K is a promising direction for improving both performance and efficiency, potentially enabling richer adaptive models. Several strategies could support dynamic K selection, such as thresholding the selector map scores or conditioning K on the input image and a desired accuracy or efficiency level.
>
> We explored one approach during the rebuttal period by selecting high-res patches based on logit thresholds rather than top-K selection (our default in the paper). For both methods, we swept values and measured accuracies on ImageNet-1K. We found that both approaches performed similarly with respect to accuracy and speed. For example, using a logit threshold of 0.03 resulted in 79.48% accuracy and 78.35 tokens on average; whereas setting K=80 tokens resulted in 79.72% accuracy. Given the broader relevance of this idea – extending beyond LookWhere to adaptive token selection more generally – we leave a deeper investigation to future work.
>
> [R1] S. Venkataramanan et al., “Franca: Nested Matryoshka Clustering for Scalable Visual Representation Learning,” arXiv preprint arXiv:2507.14137, 2025.

---

> > ### Comment · Reviewer_7H5B · 2025-08-09
> >
> > Thanks for the response. The additional experiments using Franca as a teacher model were convincing. These experiments effectively demonstrate that the proposed framework is not limited to DINOv2, and my concerns about its generalizability have been resolved.
> >
> > Your response regarding the information loss of global tokens also resolves much of my concern from a practical standpoint. The performance comparison against the full-resolution baseline provides reasonable evidence for the effectiveness of your approach. On the other hand, as this is an indirect validation, the argument for the method's soundness would be further strengthened by a more direct analysis of information loss at the token level (e.g., by comparing representation similarity). Please consider this a suggestion for future work.

---

### Official Review · Reviewer_HJW7 · 2025-06-28

**Clarity:** 3
**Significance:** 3
**Originality:** 3
**Rating:** 5
**Confidence:** 3

**Summary:**

The paper proposes LookWhere, a self-supervised method for vision transformers aiming to reduce computational cost. The method splits inference into two components: a low-resolution selector determining "where to look" by predicting regions of visual interest, and a high-resolution extractor processing only the selected regions to determine "what to see." The selector and extractor pair is jointly pretrained via distillation from a self-supervised teacher model (DINOv2), enabling efficient representation learning. Experiments demonstrate LookWhere's efficacy in reducing computational overhead while maintaining or improving performance on different benchmarks.

**Questions:**

1. How dependent is LookWhere's performance on the choice of the self-supervised teacher? Please discuss the robustness of LookWhere when trained using alternative teachers.
2. What prevents fine-tuning the selector for task-specific scenarios?
3. Could the selective mechanism make LookWhere more sensitive or robust to adversarial examples?

**Ethical Concerns:**

["NO or VERY MINOR ethics concerns only"]

**Final Justification:**

The authors' rebuttal has addressed all my concerns and questions. After reading the rebuttal and other reviewers' comments, I decide to keep my score and recommend accept.

**Limitations:**

Yes

**Paper Formatting Concerns:**

No formatting concerns

**Quality:**

3

**Strengths And Weaknesses:**

Strengths:

1. The paper uses an interesting split-resolution selector-extractor strategy, which effectively utilizes adaptive computation.
2. The paper presents an interesting joint distillation framework (what-where) from a self-supervised teacher that enhances the computational efficiency.
3. Strong experimental results showing efficiency gains.

Weaknesses:

1. Relying on a single self-supervised teacher (DINOv2) for distillation might limit the generalizability.
2. The exploration of selector fine-tuning for task-specific optimization is limited.

---

> ### Author Rebuttal · Authors · 2025-07-31
>
> # Response to Reviewer HJW7
>
> We thank the reviewer for their thoughtful, constructive and encouraging feedback. In particular, we appreciate their recognition of our strong experimental results and are grateful for their suggestions for improving our paper, which we address in this rebuttal.
>
> \
> **Robustness of LookWhere Using Alternative Teachers:** \
> LookWhere relies on DINOv2 – and any other teacher model – in two key ways: to distill where to look via its saliency maps, and what to see via its representations. While many modern models provide high-quality patch representations suitable for training the extractor, our choice of DINOv2 is primarily motivated by its highly semantic and clean attention maps, which allow the selector to identify visually salient regions without supervision from labels or tasks.
>
> Importantly, any teacher model with semantic attention maps could be used effectively within our framework. To verify LookWhere’s effectiveness using other teachers, we conducted new experiments during the rebuttal period using Franca [R1] – a new visual foundation model – with a ViT-B backbone as the teacher, keeping all other hyperparameters fixed. We note that Franca was released after our submission, and demonstrates that LookWhere can be easily applied to and benefit from future models. Franca uses a CLS token for global context and does not employ registers, yet we observed that distillation remains effective. Following Section 3.1, our LookWhere-Franca model with k = 128 (9.3% of tokens) achieves 71.2% accuracy on ImageNet-HR and 61.4% on ADE20K. Furthermore, our finetuned LookWhere-Franca model achieves 81.0% on ImageNet which compares favorably to DINOv2 (Table 1). We also finetuned this LookWhere-Franca model on downstream tasks, obtaining the following results:
> - **Traffic:** 90.2%, 91.4% and 93.4% using K values of 518 (10%), 1036 (20%), and 2592 (50%).
> - **Birds:** 86.6%, 88.1% and 88.2% using K values of 136 (10%), 273 (20%) and 684 (50%).
> - **Billiards:** 97.1%, 97.0%, and 97.0% using K values of 518 (10%), 1036 (20%), and 2592 (50%) (see “Generalization to More Domains” in our response to reviewer J83A)
>
> These results compare favorably to those obtained with DINOv2, and we will reference them alongside our other finetuning results in section 3.2 while also including them near Tables 3 and 4. We reiterate our thanks to the reviewer for this insight that broadens the generality of LookWhere; our experiments confirm its effectiveness, regardless of the teacher used.
>
> \
> **What prevents finetuning the selector for task-specific scenarios?:** \
> While nothing fundamentally prevents finetuning the selector for task-specific scenarios, at least in theory, its _discrete_ patch selection makes standard gradient descent inapplicable without modifying the (supervised) training scheme or overall model architecture. One possible direction is adapting a differentiable top-k selection method, such as DPS, or using a reinforcement learning approach as in PatchDrop, to enable selector finetuning after pretraining.
>
> Our results suggest the pretrained selector already generalizes well across tasks – evidenced by the selector maps for Traffic Sign Recognition and CUB (Figures 12 and 13), and strong performance on the newly added Billiard Ball task (see our response to reviewer J83A). In such cases, especially with a suitable teacher, we expect limited gains from selector finetuning. Nevertheless, we agree with the reviewer and acknowledge that finetuning the selector could still improve performance and enhance LookWhere’s flexibility, as well as other adaptive selection schemes such as LTRP, which is why we explicitly mention this limitation in our paper. Given our positive results regarding selector generalization, we hope our findings encourage future research into selector specialization through finetuning.
>
> \
> **Could the selective mechanism make LookWhere more sensitive or robust to adversarial examples?:** \
> We thank the reviewer for this insightful question and investigated it further during the rebuttal period by evaluating our LookWhere model under a PGD attack. We find that LookWhere significantly increases robustness to adversarial examples compared to DINOv2, often achieving an additional 20+% accuracy. We summarize these results below and will include them in a new subsection focused on robustness (given an additional room from a 10th page).
>
> **PGD attacks with epsilon=1/255**
> - **LookWhere-Base**
>     - **3 steps:** 32.1% acc. on ImageNet-Val
>     - **5 steps:** 27.5% acc. on ImageNet-Val
>     - **10 steps:** 24.8% acc. on ImageNet-Val
> - **DINOv2-Base**
>     - **3 steps:** 8.9% acc. on ImageNet-Val
>     - **5 steps:** 6.5% acc. on ImageNet-Val
>     - **10 steps:** 5.3% acc. on ImageNet-Val
>
> [R1] S. Venkataramanan et al., “Franca: Nested Matryoshka Clustering for Scalable Visual Representation Learning,” arXiv preprint arXiv:2507.14137, 2025.

---

> > ### Comment · Reviewer_HJW7 · 2025-08-07
> >
> > Thanks for the response. I think the respnse has resolved my concerns and I will keep my positive score.

---

### Official Review · Reviewer_HzUw · 2025-06-30

**Clarity:** 3
**Significance:** 3
**Originality:** 3
**Rating:** 4
**Confidence:** 3

**Summary:**

This paper proposed a self-supervised learning based distillation framework comprised of a selector for processing low-resolution images, a FFN to ‘upscale’ attention map, an extractor for tokenization and a task-specific predictor for downstream fine-tuning. And then authors conducted experiments on ImageNet-1K, Caltech-UCSD Birds and Traffic Signs dataset for image classification and ADE20K for semantic segmentation to evaluate proposed method’s effectiveness and efficiency.

**Questions:**

1. In this paper, authors initialized the selector and extractor with the **first** $L_{low}$ layer from the teacher model DINOv2 and provided an ablation of layer depth in `Fig.7.a`. What about the layer location’s effect on distillation(e.g., 3 layers of 1-3, 4-6, 7-9, or 9-12)? There are some previous research revealing that bottom layer gains more semantic information than top layers, which may bring more gains for distillation.
2. Could this proposed distillation method also be applied on other architectures beyond ViT? After the process of selector, top-k tokens may loss its 2D structure, which may bring a limitation of the extractor. While the architecture of selector that process full low-resolution images has more choices beyond ViT. Could author make a try of more architectures?

**Ethical Concerns:**

["NO or VERY MINOR ethics concerns only"]

**Final Justification:**

Authors provided detailed responses and additional experiment results on alternative selector initialization schemes, which have addressed most of my previous concerns.

**Limitations:**

See `Weakness` and `Question` part.

**Quality:**

3

**Strengths And Weaknesses:**

Strengths:
1. Well-motivated: The authors comprehensively introduced the motivation of each section or design.
2. Clear writing: This paper offers a clear and logical structure that is easy to follow.

Weaknesses:
1. About `Fig.2`’s demonstration: According to the `Sec.2.1`, this figure may loss some key details about FFN for high-resolution selector map, which also makes the $L_{map}$ a little confusing.
2. According to the ablation in `Fig.7.c`, the register token brings large performance gain on both cls and seg task. Would the effectiveness of distillation still exist when other models (e.g. contrastive learning based CLIP or reconstruction based MAE) without register token are used as teacher model?
3. This paper provided many metrics(e.g. FLOPS, memory, IPS) for efficiency comparison. Could author additionally provide the whole ViT-S or -B based model parameters information (including FFN for high-resolution selector map) for a more comprehensive with compared methods in `Tab.1-2`, `Fig.4-5`.

I'd be more than willing to revise my score based on the author's further responses.

---

> ### Author Rebuttal · Authors · 2025-07-31
>
> # Response to Reviewer HzUw
>
> We thank the reviewer for their thoughtful feedback, including their recognition of our clear writing and well-motivated approach. We reply to address concerns regarding effectiveness without registers, in addition to new experimental results investigating layer initialization and LookWhere’s generalization to other architectures.
>
> \
> **Does the effectiveness of distillation exist when the teacher lacks registers?:** \
> We thank the reviewer for this thoughtful question and agree that registers are indeed beneficial for distillation. However, based on experiments conducted during the rebuttal period, we confirm that registers are not essential for LookWhere’s effectiveness. Registers offer two key benefits: (i) they pass global context from the low-resolution image via the selector, and (ii) they typically produce cleaner attention maps with fewer artifacts, which helps the selector learn better maps. However, these benefits are not unique to registers. Global context can also be delivered through mechanisms such as a CLS token or pooling patches, and teachers without registers can still produce semantic attention maps.
>
> To verify LookWhere’s effectiveness in the absence of registers, we conducted new experiments during the rebuttal period using Franca [R1] – a new visual foundation model – with a ViT-B backbone as the teacher, keeping all other hyperparameters fixed. We note that Franca was released after our submission, and demonstrates that LookWhere can be easily applied to and benefit from future models. Franca uses a CLS token for global context and does not employ registers, yet we observed that distillation remains effective. Following Section 3.1, our LookWhere-Franca model with k = 128 (9.3% of tokens) achieves 71.2% accuracy on ImageNet-HR and 61.4% on ADE20K. Furthermore, our finetuned LookWhere-Franca model achieves 81.0% on ImageNet which compares favorably to DINOv2 (Table 1). We also finetuned this LookWhere-Franca model on downstream tasks, obtaining the following results:
> - **Traffic:** 90.2%, 91.4% and 93.4% using K values of 518 (10%), 1036 (20%), and 2592 (50%).
> - **Birds:** 86.6%, 88.1% and 88.2% using K values of 136 (10%), 273 (20%) and 684 (50%).
> - **Billiards:** 97.1%, 97.0%, and 97.0% using K values of 518 (10%), 1036 (20%), and 2592 (50%) (see "Generalization to More Domains" in our response to reviewer  J83A)
>
> These results compare favorably to those obtained with DINOv2, and we will reference them alongside our other finetuning results in section 3.2 while also including them near Tables 3 and 4. We reiterate our thanks to the reviewer for this insight that broadens the generality of LookWhere; our experiments confirm its effectiveness, regardless of whether the teacher has registers.
>
> \
> **Selector and Extractor Layer Initialization:** \
> We want to clarify that only the selector is initialized with the first $L_{\text{low}}$ layers from the teacher; the extractor retains all layers. While a similar strategy could be applied to the extractor, our primary goal is to reduce the selector’s depth for improved efficiency. We retain the extractor’s depth to maintain performance while ensuring efficiency by processing fewer tokens.
>
> Following the reviewer’s suggestion, we explored alternative selector initialization schemes during the rebuttal period. Specifically, we trained four ViT-S models for 100 epochs each using a learning rate of 0.0002, and initializing the selector with different sets of teacher layers: {[0, 1, 2], [3, 4, 5], [6, 7, 8], [9, 10, 11]}. We summarize these results in Tables T1 and T2 below and generally find LookWhere to be robust to layer initialization – especially with larger K values. At smaller values of K, LookWhere benefits slightly from initializing with deeper layers, aligning with the reviewer’s insight. We thank the reviewer for this suggestion and will include these results alongside our other ablations in Figure 7.
>
> **Table T1. Layer Initialization ImageNet kNN Accuracy (%) for Varying Number of Patches (K)**
>
> | Layer Initialization | K=16   | 72   | 128  | 256  |
> |----------------------|------|------|------|------|
> | 0-1-2                | 38.52| 55.90| 62.76| 68.68|
> | 3-4-5                | 39.88| 56.68| 62.20| 68.36|
> | 6-7-8                | 39.90| 56.66| 62.42| 68.44|
> | 9-10-11              | 40.54| 57.32| 63.02| 68.34|
>
> **Table T2. Layer Initialization ADE kNN Accuracy (%) for Varying Number of Patches (K)**
>
> | Layer Initialization |  K=16   | 72   | 128  | 256  |
> |----------------------|------|------|------|------|
> | 0-1-2                | 48.10| 57.88| 60.33| 63.81|
> | 3-4-5                | 47.78| 57.58| 60.45| 64.31|
> | 6-7-8                | 48.06| 57.77| 60.24| 64.24|
> | 9-10-11              | 48.49| 57.68| 60.61| 64.32|
>
> \
> **Could the proposed distillation method be applied beyond ViTs?:**
>
> Yes, LookWhere can generalize beyond ViTs, though different considerations are needed for the extractor and selector architectures.
>
> The extractor relies on a partitioning of the input, typically into tokens, that enables: i) selection and processing of sparse inputs, and ii) distillation using partial inputs to learn semantic representations. These two requirements are satisfied by patchification of the high-resolution input, regardless of how the selected patches are later processed. For example, both DPS and IPS use CNN-based selectors and explore different extraction methods, such as pooling or shallow transformers. While alternative extractor architectures are viable, we focus on ViTs due to their effectiveness at seamlessly processing sparse inputs.
>
> Regarding 2D structure, we appreciate the reviewer’s concern that selecting the top-k tokens may disrupt spatial coherence. However, we emphasize that the selector is explicitly trained to select the most informative tokens, meaning any dropped tokens are less critical to the image representation. During pretraining, LookWhere’s patch and class-token distillation losses help ensure that full representation of the high-resolution input can be approximated using only sparse high-resolution tokens which is aided by 2D positional embeddings, also helping ensure structure is maintained.
>
> The selector plays a simpler role of approximating the teacher’s saliency map, which we realize through attention. Hence, any architecture capable of image processing is suitable here. To validate this, we conducted additional experiments during the rebuttal period, training two ViT-B models for 100 epochs each: one using a DINOv2-initialized selector and the other using a CNN initialized EfficientNetV2 and summarize kNN results on ImageNet-HR classification and ADE20K segmentation below (akin to Tables 1 and 2 from our paper) using K values of 16 (1.2%), 72 (5.3%), and 128 (9.3%):
> - **LookWhere w/ EfficientNetV2 Selector**
>     - **ImageNet-HR:** 71.84%, 81.18% and 82.48%
>     - **ADE20K:** 49.34%, 58.80% and 62.25%
> - **LookWhere w/ DINOv2 Selector**
>     - **ImageNet-HR:** 66.16%, 74.92% and 78.42%
>     - **ADE20K:** 49.34%, 58.80% and 62.25%
>
> We also finetuned LookWhere with an EfficientNetV2 Selector for 10 epochs on ImageNet and obtained top-1 accuracy of 82.3%. Overall, we confirm EfficientNet comparable performance to DINOv2 as a selector across all these experiments and validate the ability of LookWhere to generalize beyond ViTs. We will add a new subsection referencing these results, and discussing alternative architectures, to our experiments should our paper be accepted.
>
> \
> **Figure 2’s demonstration:** \
> We appreciate the reviewer’s concern regarding the clarity of our central figure. We intentionally abstract away the internals of both the selector (including the FFN) and the extractor to simplify the depiction of LookWhere’s computational flow. We aim to avoid tying the diagram to specific implementation details. While our paper focuses on a ViT-based instantiation of LookWhere, the framework is not limited to ViTs. For example, as demonstrated above, the selector can also be implemented using a CNN. We will revise our figure, clarifying that the output of the selector is a _high-resolution_ selector map to ease interpretation of the losses.
>
> \
> **ViT-S or -B based model parameters information:** \
> We agree that including parameter counts for all models, including baselines, facilitates fair comparison.
> DINOv2, DTEM, PiToME, and ATC are all plain ViT-B architectures with 86 million parameters. In addition to a plain ViT-B, LTRP adds a 22-million-parameter ViT-Small ranking model using 108 million parameters in total. In addition to a plain ViT-B, LookWhere adds 3/12 of a ViT-B along with an FFN to realize its selector, using 109.8 million parameters. However, we measured total inference memory usage across all models to be approximately equal.
>
> [R1] S. Venkataramanan et al., “Franca: Nested Matryoshka Clustering for Scalable Visual Representation Learning,” arXiv preprint arXiv:2507.14137, 2025.

---

> > ### Comment · Reviewer_HzUw · 2025-08-06
> >
> > Thanks for authors' detailed responses and additional experiment results, which have addressed most of my previous concerns. I will keep my positive score and recommend accept.

---

### Official Review · Reviewer_J83A · 2025-07-03

**Clarity:** 3
**Significance:** 2
**Originality:** 2
**Rating:** 5
**Confidence:** 4

**Summary:**

The authors propose LookWhere, a method for efficient processing of high-resolution images with vision transformers. LookWhere consists of two models, the selector and the extractor. The selector processes the input image in low resolution to decide which image patches (tokens) should be processed in higher resolution. Then, the extractor receives as input only these patches from the high resolution image, and outputs the final representation that can be used for downstream tasks. The main source of efficiency is the described selective processing, since only a small part of the input image is processed in high resolution.

The selector and extractor models are trained together end-to-end, by distilling an existing self-supervised teacher, i.e., DINOv2. The authors use 3 simple loss terms, (1) MSE between the class token of the extractor and that of the teacher, (2) MSE between the patch tokens of the extractor and those of the teacher, (3) KL divergence between the output of the selector (it is an attention map over the patch sequence of the full resolution image), and a combination of attention maps from the teacher.

The authors test their model on image classification and semantic segmentation, by finetuning the extractor on the respective datasets along with a task-specific head; the selector is expected to generalize after distillation, and is not finetuned for different tasks. Baselines consist of other adaptive computation methods, and DINOv2 for reference, since it is the teacher model. The authors first test the accuracy and efficiency of their method on ImageNet-1K and ADE20K, which are not of particularly high resolution, but are well established benchmarks. Then they test on Traffic Signs dataset with images of resolution $994 \times 994$ px, where the regions of interest (signs) are relatively small. The last dataset is CUB Birds, which again is not of particularly high resolution, but requires fine-grained classification, so it can test a different capability of the model. Overall, LookWhere compares favorably to the adaptive baselines, while DINOv2 usually has the best performance, but is also considerably slower. Finally, the authors conduct a number of ablations to justify their design choices.

**Questions:**

My main question is why the authors didn’t explore the compute-performance tradeoff that LookWhere could provide? Especially since DINOv2 achieved better performance in all but the Traffic Signs dataset, indicating that there is value in processing more image patches.

**Ethical Concerns:**

["NO or VERY MINOR ethics concerns only"]

**Final Justification:**

The authors provided additional experiments during the rebuttal period, showing the efficacy of LookWhere in a wider range of datasets. i.e., AID, NIH Chest X-ray dataset, and Billiard Ball Dataset. In addition, they trained LookWhere with a wider range of $k$ values, addressing my main concern regarding the ability of LookWhere to trade off compute for performance and vice versa. LookWhere was able to perform comparably to its Teacher in a number of datasets for less compute, demonstrating its ability to utilize different $k$ when appropriately trained, and its potential to further improve in light of a stronger Teacher model. For these reasons, I increased my score from $3$ to $5$.

**Limitations:**

yes

**Paper Formatting Concerns:**

I didn't notice any formatting issues.

**Quality:**

3

**Strengths And Weaknesses:**

- Originality:
1. The idea of selective processing by decoupling “where” and “what”, along with using a low resolution image for guidance, are not novel.
2. However, the proposed method has simple and straightforward losses, which I assume is good for training stability, and takes advantage of modern powerful models through distillation.
- Quality:
1. I can see both strengths and weaknesses related to the experiments. On the strengths side, I think the authors use appropriate baselines, and they test on both classification and segmentation, which is not a common task for adaptive models. In addition, as I described in the previous Section, different experiments test for different capabilities, for example ImageNet-1K contributes as a popular benchmark, while Traffic Signs can test the attention policy of the selector.
2. Another strength of the paper is the ability of the selector to generalize across datasets without finetuning.
3. On the weaknesses side, I found surprising that the authors did not provide compute-performance trade-off curves by varying $k$, which determines the number of high resolution patches processed by the extractor. In general, I think it is important for an adaptive model to be able to trade off compute for performance and vice versa, so it can adjust to scarcity of resources or increased performance demands. The authors vary $k$ only in the segmentation experiments in Table 2, and in a brief ablation experiment in Fig. 7 (d), however, I think it would be appropriate to test different values of $k$ in all experiments. For example, in the fine-grained classification experiments in Fig. 4, we see that LookWhere strikes a good balance between throughput and accuracy, but it's neither the fastest nor the most accurate, so, I think it would strengthen the results if there was a throughput-accuracy trade-off curve to show that LookWhere can beat all baselines for comparable throughput (increase $k$ to beat more accurate baselines and decrease $k$ to beat faster baselines at similar throughput). I see that in Tables 3 and 4 in the Appendix, the authors provide LookWhere results for different $k$, but LookWhere never reaches the accuracy of the top performing baselines, while it remains considerably faster, so, I think higher values of $k$ could be used; for example, in Table 4 in the Appendix, LookWhere has about 1% lower accuracy compared to DTEM, but its eval speed is about $\times 5$.
5. I think the authors could have used more diverse high-resolution datasets, e.g., satellite or medical images.
6. I think the parameters of all models, including the baselines, should be provided in at least one of the Tables, e.g., Table 1, for easy comparison.
- Clarity:
1. The paper is very well written, and easy to follow. The authors provide a lot of implementation details, and a lot of ablation studies, which can be very useful for people who are interested to build on this work in the future.
2. I only found Fig. 5 a bit confusing, what do the authors mean by cumulative FLOPs?
3. Minor typos, e.g., ln 263: “resolution a 3 layers” should be “and”.
- Significance:
1. Overall, it is a very well written paper, with carefully designed experiments, however, given that the originality of the ideas is limited, I think the results should have been stronger.

---

> ### Author Rebuttal · Authors · 2025-07-31
>
> # Response to Reviewer J83A
>
> We thank the reviewer for their thoughtful feedback, particularly their recognition of our extensive ablations, carefully chosen baselines, and clear presentation. We reply to clarify the originality of our work and provide additional ablations as requested, focusing on LookWhere’s compute-performance tradeoff and its generalization to more diverse, high-resolution datasets.
>
> \
> **Originality of LookWhere:** \
> While we agree that decoupling “where” and “what” in selective processing is not novel, we believe LookWhere introduces a key innovation through its scalable and straightforward joint pretraining of the selector and extractor via distillation; simultaneously learning where to attend from attention and what to extract from representation, unlike all prior work. As the reviewer notes, it is precisely this distillation that enables LookWhere’s simple yet effective losses, promoting stable and efficient training while leveraging the capabilities of powerful modern models.
>
> In the related work section, we explain LookWhere’s relationship to prior token selection methods such as IPS, DPS, and PatchDrop, which also adopt a decoupled approach. However, we emphasize that these methods rely on complex, task-specific, or multi-stage training pipelines. In contrast, LookWhere achieves superior efficiency and performance through a single-pass distillation process that pretrains both the selector and extractor jointly. By leveraging self-supervised representations, we find the selector generalizes across tasks, requiring only the extractor to be finetuned. We appreciate the reviewer’s feedback and will further underline our novelty by this dual distillation in the revised related work section.
>
> \
> **Generalization to More Domains:** \
> We thank the reviewer for suggesting an evaluation on more diverse datasets and fully agree that doing so strengthens our generalization claims. During the rebuttal period, we extended our evaluation to three additional datasets. Following our established methodology, we finetuned both LookWhere and DINOv2 on these new downstream tasks. The results continue to demonstrate LookWhere’s strong performance, as summarized below:
>
> - **Aerial Image Dataset (AID) [R1]** of 10K images for aerial scene classification across 30 classes. Images were resized from 600px to 602px to match DINOv2’s patch size. We used a 50/50 train/test split and finetuned for 30 epochs. We measure top-1 accuracies of 97.4%, 98.0%, and 98.3% using K values of 184 (10%), 369 (20%), and 924 (50%), respectively, which continues to be competitive with DINOv2’s 98.7% when processing _all_ patches.
>
> - **NIH Chest X-ray Dataset [R2]** of 112K medical images (90K train / 22K test) for multi-label clinical diagnosis across 14 labels. Images were resized from 1024px to 1022px and finetuned for 10 epochs. Following prior work [R3, R4], we measure AUC-ROC scores of 78.7%, 79.6%, and 81.1% using K values of 532 (10%), 1065 (20%), and 2664 (50%), respectively. DINOv2 achieves a comparable AUC-ROC of 82.2% while processing _all_ patches.
>
> - **Billiard Ball Dataset [R5]** of 18K images (8K train / 10K test) used to evaluate interpatch reasoning. Each image contains 4-8 balls numbered 1-9; the task is to identify the larger number between the left-most and right-most balls. Images were resized from 1000px to 1008px and finetuned for 30 epochs. We measure top-1 accuracies of 97.3%, 97.5%, and 97.4% using K values of 518 (10%), 1036 (20%), and 2592 (50%), respectively. LookWhere achieves nearly the same accuracy as DINOv2 (97.6%) while only processing _a fifth_ of the patches.
>
> \
> **LookWhere’s Compute Performance Tradeoff:** \
> We agree that evaluating LookWhere’s ability to trade off compute for performance is important. As the reviewer notes, we already explored this in two settings: segmentation (Table 2) and ImageNet classification (Figure 7d). During this rebuttal, we vary K more extensively on AID, NIH Chest X-ray, Billiard Balls, and CUB (see below) and will plot accuracy-throughput tradeoff curves, similar to Figure 7d, for these experiments in our paper, should it be accepted.
>
> We initially observed no performance gains using our original LookWhere model during finetuning on CUB across a wider K range, beyond the ratio of 0.2 already included in our paper. This lack of improvement is likely due to the model’s pretraining, which sampled 16 - 128 tokens (out of 1369). To explore this further, we pretrained another model during the rebuttal period using a wider range of token samples (256 - 512) to better support larger K during finetuning.
>
> We then finetuned this model on CUB using our standard approach and measured top-1 accuracies of 88.7%, 90.1% and 90.4% using K values of 136 (10%), 273 (20%) and 684 (50%), respectively. This updated model matches DINOv2’s performance of 90.4% using only half the patches, outperforming all baselines except DTEM, and coming within 0.6% of DTEM’s accuracy while reducing FLOPs by 4.42x. These results further support LookWhere’s ability to reduce computation cost with minimal performance loss. We also note that LookWhere’s performance is closely tied to that of the teacher model, and could improve further with more powerful teachers.
>
> \
> **Parameters of all Models:** \
> We agree that including parameter counts for all models, including baselines, facilitates fair comparison.
> DINOv2, DTEM, PiToME, and ATC are all plain ViT-B architectures with 86 million parameters. In addition to a plain ViT-B, LTRP adds a 22-million-parameter ViT-Small ranking model using 108 million parameters in total. In addition to a plain ViT-B, LookWhere adds 3/12 of a ViT-B along with an FFN to realize its selector, using 109.8 million parameters in total. However, we measured total inference memory usage across all models to be approximately equal.
>
> \
> **Cumulative FLOPs:** \
> In Figure 5, “cumulative FLOPs” refers to the total FLOPs consumed during training up to a given point. For example, if a forward and backward pass over a batch requires X FLOPs and there are Y batches per epoch, the cumulative FLOPs after the first epoch would be Y * X, after the second epoch 2 * Y * X, and so on. We will provide further clarification in Figure 5’s caption.
>
> \
> **Minor typos:** \
> Thank you for pointing this out. We will correct the typo in the final version.
>
> [R1] G.-S. Xia et al., “AID: A benchmark data set for performance evaluation of aerial scene classification,” IEEE Transactions on Geoscience and Remote Sensing, vol. 55, no. 7, pp. 3965–3981, 2017.
>
> [R2] X. Wang, Y. Peng, L. Lu, Z. Lu, M. Bagheri, and R. M. Summers, “Chestx-ray8: Hospital-scale chest x-ray database and benchmarks on weakly-supervised classification and localization of common thorax diseases,” in Proceedings of the IEEE conference on computer vision and pattern recognition, 2017, pp. 2097–2106.
>
> [R3] A. Kaku, S. Upadhya, and N. Razavian, “Intermediate layers matter in momentum contrastive self supervised learning,” Advances in Neural Information Processing Systems, vol. 34, pp. 24063–24074, 2021.
>
> [R4] E. Tiu, E. Talius, P. Patel, C. P. Langlotz, A. Y. Ng, and P. Rajpurkar, “Expert-level detection of pathologies from unannotated chest X-ray images via self-supervised learning,” Nature biomedical engineering, vol. 6, no. 12, pp. 1399–1406, 2022.
>
> [R5] J.-B. Cordonnier, A. Mahendran, A. Dosovitskiy, D. Weissenborn, J. Uszkoreit, and T. Unterthiner, “Differentiable patch selection for image recognition,” in Proceedings of the IEEE/CVF Conference on Computer Vision and Pattern Recognition, 2021, pp. 2351–2360.

---

> ### Comment · Reviewer_J83A · 2025-08-04
> **Response to Authors' Rebuttal**
>
> I would like to thank the authors for their detailed response, and for conducting additional experiments. As I mentioned in my initial review, my main concern had to do with the compute-performance trade off curves, which the authors address by conducting additional experiments on CUB, on new datasets, and by mentioning that they will include tradeoff curves in all experiments. I find it very encouraging that LookWhere matched DINOv2 performance with half the patches on CUB, so, I am willing to increase my score, though, I have some additional questions:
> - What's the actual speed up of LookWhere compared to DINOv2 when processing half the patches on CUB?
> - How adaptive baselines compare against LookWhere at the newly added datasets?
> - About Fig. 5, I thank the authors for their clarification, an additional question I have, is what's the intervals between the markers, e.g., 1 epoch?

---

> > ### Author Response · Authors · 2025-08-05
> >
> > We thank the reviewer for continuing the discussion and are happy to address these questions.
> >
> > - With half the total patches, LookWhere increases speed by ~2.5x and reduces FLOPs by ~2.2x in both training and inference compared to DINOv2 (which receives all patches)
> > - The adaptive baselines are very slow to train, e.g., ATC and PiToMe are 185 and 13 times slower, respectively, than LookWhere on 1k by 1k px images, and DTEM goes OOM on a 24 GB VRAM GPU. Furthermore, DINOv2 with all patches is the approximate upper bound for accuracy in this setup. Thus we prioritized LookWhere and DINOv2 training runs during the rebuttal period. We were able to run adaptive baselines for Billiard Balls only: PiToMe and ATC achieved ~23%, DTEM went OOM, IPS achieved 90.4%, and DPS achieved 69.1%. In our camera-ready version, we will include all baselines methods for these 3 new datasets. We again thank the reviewer for suggesting these experiments — we agree they will strengthen our paper.
> > - The interval between the markers is 5 epochs. Thank you, we will include it in a revised caption.

---

> > > ### Comment · Reviewer_J83A · 2025-08-05
> > >
> > > I thank the authors for their response. I think that the described additional results show a broader applicability of LookWhere, since its efficacy is demonstrated in a wider range of datasets. In addition, LookWhere can offer a more adjustable compute-performance trade-off when trained with a bigger range of $k$ values, and it can match the performance of its teacher for less compute, showing that a stronger teacher may further boost LookWhere performance. Based on these, I will increase my score accordingly.

---

### Author Response · Authors · 2025-08-05
**Summary of Rebuttal**

We thank the reviewers once more for their feedback and thank Reviewer J83A for already engaging in the discussion. We would like to summarize our responses and are happy to engage with these or other points.

**Other Teachers.** The most common question asked if LookWhere could apply to teachers beyond DINOv2 (reviewers HzUw, HJW7, and 7H5B). In response, we demonstrated that LookWhere can leverage a Franca teacher, which is a SOTA vision foundation model made public at the start of the rebuttal period. In our experiments our LookWhere-Franca model performed well across sparse high-resolution datasets and ImageNet. Thus our framework can be quickly leveraged to approximate the computation of new vision foundation models.

**Varying K.** Reviewers J83A and 7H5B asked about varying K, i.e., the number of high-res patches the extractor processes to trade off accuracy and cost. In our submission, we varied K for ImageNet and ADE20K. In our response, we varied K for all other datasets (including datasets added in response to other questions). We also experimented with a dynamic K strategy, which achieved comparable results to static K but demonstrates that a simple thresholding of the selector predictions is possible for adaptivity. We cannot plot more curves of performance across K due to NeurIPS rules, however we will add these plots to our camera-ready version if the paper is accepted.

**More High-Res Datasets**. Reviewer J83A asked for more high-res domains, e.g., satellite or medical images. In response, we demonstrated that LookWhere generalizes well to the Aerial Image Dataset (remote sensing), NIH Chest X-Rays (medical), and the Billiards Balls dataset [S1] (made specifically to evaluate sparse high-res visual recognition).

We remain available if there are any more points to address during the discussion phase and thank the reviewers for their time.

[S1] J.-B. Cordonnier, A. Mahendran, A. Dosovitskiy, D. Weissenborn, J. Uszkoreit, and T. Unterthiner, “Differentiable patch selection for image recognition,” in Proceedings of the IEEE/CVF Conference on Computer Vision and Pattern Recognition, 2021, pp. 2351–2360.

---

### Note · Authors · 2025-08-13

We sincerely thank the reviewers for their thoughtful feedback and the AC for their consideration of our paper.

The most common concern from reviewers (HzUw, HJW7, and 7H5B) was whether LookWhere could be applied to teachers beyond DINOv2. In our rebuttals, we demonstrated that LookWhere works well with a Franca teacher (a new SOTA foundation model released at the start of the rebuttal period). All reviewers with this concern acknowledged our demonstration, most explicitly reviewer 7H5B, who said this question was resolved.

We were also asked to vary K (the number of high-res patches the extractor processes) in more settings by reviewer J83A. In our rebuttals, we varied K for all datasets that were missing in our submission and for additional datasets. Reviewer J83A stated it resolved their concern. Reviewer J83A also requested LookWhere demonstrations on additional high-resolution datasets, specifically those related to medical or remote sensing. In response, we demonstrated that LookHere performs well on the Aerial Image Dataset (remote sensing), NIH Chest X-Rays (medical), and the Billiards Balls dataset — this satisfied reviewer J83A, who agreed to raise their score.

We look forward to incorporating these additional experiments into the paper, along with other points that were brought up by reviewers, in our camera-ready paper (should our submission be accepted).

---

### Decision · Program_Chairs · 2025-09-17

**Decision:**

Accept (poster)

**Comment:**

This paper introduces LookWhere, a method for building inference-efficient vision transformer. It consists of a low-resolution selector (to identify informative regions) and a high-resolution extractor (to process the patches). The approach is trained by distilling self-supervised models and achieves strong accuracy-efficiency trade-offs across several benchmarks. The reviewers generally praised the paper in their initial reviews, and the authors successfully answered all of the reviewers concerns, strengthening the claims. The additional evidence (more datasets and more teacher models) made reviewers improve their scores. Overall this is a good paper and I recommend acceptance as a poster.